# AFD thermosensory neurons mediate tactile-dependent locomotion modulation in *C. elegans*

Manuel Rosero[1,2], Jihong Bai[1,2]*

[1]Basic Sciences Division, Fred Hutchinson Cancer Center, Seattle, United States; [2]Molecular and Cellular Biology Graduate Program, University of Washington, Seattle, United States

## eLife Assessment

The manuscript presents **important** findings on how *C. elegans* can utilize distinct molecular mechanisms and circuit engagements to regulate tactile-dependent locomotory behaviours through the AFD thermosensory neuron. The authors use multiple techniques including microfluidics, genetic manipulations and single-copy rescue experiments, to provide **compelling** evidence for the role of AFD/AIB electrical synaptic connections in this behaviour. The reviewers are satisfied with the comprehensive revisions made by the authors.

*For correspondence:
jbai@fredhutch.org

**Competing interest:** The authors declare that no competing interests exist.

## Abstract

Sensory neurons drive animal behaviors by detecting environmental stimuli and relaying information to downstream circuits. Beyond their primary roles in sensing, these neurons often form additional synaptic connections outside their main sensory modality, suggesting broader contributions to behavior modulation. Here, we uncover an unexpected role for the thermosensory neuron AFD in coupling tactile experience to locomotion modulation in *Caenorhabditis elegans*. We show that while AFD employs cyclic guanosine monophosphate (cGMP) signaling for both thermotaxis and tactile-dependent modulation, the specific molecular components of the cGMP pathway differ between these two processes. Interestingly, disrupting the dendritic sensory apparatus of AFD, which is essential for thermotaxis, does not impair tactile-based locomotion modulation, indicating that AFD can mediate tactile-dependent behavior independently of its thermosensory apparatus. In contrast, ablating the AFD neuron eliminates tactile-dependent modulation, pointing to an essential role for AFD itself, rather than its sensory dendritic endings. Further, we find tactile-dependent modulation requires the AIB interneuron, which connects AFD to touch circuits via electrical synapses. Removing innexins expressed in AFD and AIB abolishes this modulation, while re-establishing AFD–AIB connections with engineered electrical synapses restores it. Collectively, these findings uncover a previously unrecognized function of AFD beyond thermosensation, highlighting its influence on context-dependent neuroplasticity and behavioral modulation through broader circuit connectivity.

## Introduction

Adaptive behavioral responses require animals to interpret sensory signals in the context of prior experiences, enabling effective responses to changing environmental conditions. For example, rodents associate contextual cues, such as odors and colors, with rewards or punishments, leading to context-dependent adjustments in approach or avoidance behaviors (*Navarro-Sánchez et al., 2024*; *Maren et al., 2013*). Disruptions in the biological processes underlying this plasticity, as seen

in neurological and psychiatric disorders, often result in impaired behavior modulation based on prior sensory experiences (*Battaglia et al., 2018*; *Leitman et al., 2011*; *Hadjikhani et al., 2009*; *Kipps et al., 2009*; *Greimel et al., 2012*). Recognizing the critical role of context-dependent behavioral modulation for survival and well-being, ongoing research focuses on uncovering the neuroplastic mechanisms responsible for these adaptive processes.

Studies across diverse organisms reveal that animals, regardless of nervous system complexity, integrate prior sensory experiences to shape behavior (*Badel et al., 2016*; *Haberkern et al., 2019*; *Sheintuch et al., 2020*; *Zars and Zars, 2006*; *Kishimoto et al., 2019*; *Sasaki et al., 2006*; *Billeter et al., 2012*; *Wolfin et al., 2018*). In the nematode *Caenorhabditis elegans*, which has only 302 neurons, locomotion strategies are modified by prior exposure to stimuli such as odors, touch, light, and starvation, demonstrating robust adaptive behaviors (*Sawin et al., 2000*; *Gourgou et al., 2021*; *Dillon et al., 2016*; *Law et al., 2004*; *Kindt et al., 2007*; *Goetting et al., 2018*; *Susoy et al., 2021*; *Ghosh et al., 2016*). The wide range of adaptive behaviors, combined with the well-mapped *C. elegans* nervous system and abundant genetic tools, have empowered the discovery of fundamental circuit mechanisms and molecular pathways underlying context-dependent behavioral modulation (*Zhen and Samuel, 2015*; *Emmons, 2018*; *Meisel and Kim, 2014*; *Metaxakis et al., 2018*; *Stout et al., 2014*).

The *C. elegans* sensory system comprises 60 ciliated sensory neurons specialized in detecting chemical, thermal, mechanical, and olfactory stimuli (*Ward et al., 1975*; *Ware et al., 1975*). Each sensory neuron has a dendritic apparatus that converts specific environmental signals into neuronal activity, which drives signal flow through downstream networks of interneurons and motor neurons, ultimately shaping behavior (*Ghosh et al., 2017*; *Hobert, 2003*; *Komuniecki et al., 2014*). Among these, AFD, a pair of bilaterally symmetric left–right neurons, serves as the primary thermosensory neurons required for thermotaxis (movement toward a preferred temperature) (*Goodman and Sengupta, 2018*). The dendritic endings of AFD, enriched with microvilli, serve as the primary site for thermal detection. Even when detached from the cell body, these dendritic endings retain calcium responsiveness to temperature stimuli (*Clark et al., 2006*). Moreover, genetically disrupting these sensory endings abolishes the temperature-dependent $Ca^{2+}$ response in AFD cell bodies and eliminates thermotaxis, underscoring their essential role in thermal sensing (*Yoshida et al., 2016*; *Singhvi et al., 2016*; *Satterlee et al., 2001*).

The second messenger cyclic guanosine monophosphate (cGMP) plays a central role in modulating AFD thermosensory activity. Temperature changes in AFD neurons trigger intracellular $Ca^{2+}$ dynamics, with $Ca^{2+}$ levels rising during warming and falling during cooling (*Clark et al., 2006*; *Aoki et al., 2022*; *Ramot et al., 2008*). This response relies on cGMP synthesis mediated by the guanylyl cyclases (GCs) *gcy-8*, *gcy-18*, and *gcy-23*, which have overlapping but distinct roles. Specifically, *gcy-8* is essential for generating thermally evoked $Ca^{2+}$ responses in AFD (*Wang et al., 2013*; *Takeishi et al., 2016*), whereas *gcy-18* and *gcy-23* act synergistically to sustain temperature sensing (*Wang et al., 2013*; *Inada et al., 2006*). The cGMP produced by these cyclases activates cyclic nucleotide-gated (CNG) channels, comprised of α-subunit TAX-4 and β-subunit TAX-2, resulting in increases in intracellular $Ca^{2+}$. Mutations in either *tax-4* or *tax-2* abolish thermally evoked $Ca^{2+}$ responses in AFD and disrupt thermotaxis, underscoring the importance of cGMP signaling for AFD-mediated thermosensation (*Ramot et al., 2008*).

AFD transmits thermal signals to several downstream interneurons, including AIY, AIZ, and AIB, which process this information to guide temperature-dependent behaviors. Thermotaxis in *C. elegans* encompasses three categories: movement toward warmer temperatures (thermophilic), movement toward cooler temperatures (cryophilic), and movement within a preferred temperature range (isothermal) (*Hedgecock and Russell, 1975*; *Ryu and Samuel, 2002*). Each behavior type depends on distinct AFD synaptic partners: AIY is critical for thermophilic and isothermal behaviors (*Kuhara et al., 2011*; *Mori and Ohshima, 1995*); AIZ is specific for cryophilic behavior (*Kuhara and Mori, 2006*; *Hobert et al., 1998*; *Chung et al., 2006*); and AIB also supports thermophilic behavior, though its role is less critical than that of AIY (*Ikeda et al., 2020*). These findings demonstrate the functional specificity of synaptic partners of AFD neurons in shaping distinct temperature-dependent behaviors.

In this study, we demonstrate that AFD plays a novel role in context-dependent locomotion modulation mediated by tactile experience, thereby expanding its function beyond temperature sensing. The mechanisms underlying this role differ markedly from those involved in thermosensation. First,

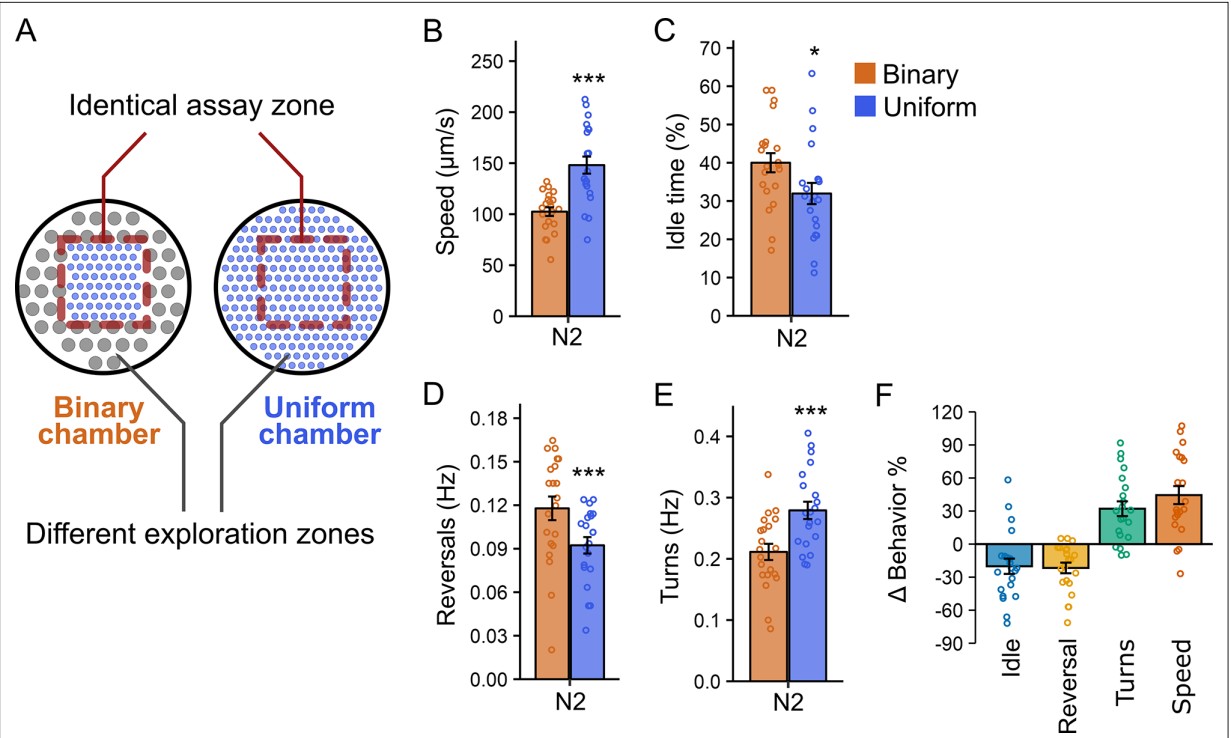

**Figure 1.** *C. elegans* displays distinct locomotion in identical environments after exploring different surrounding areas. (**A**) Schematic of two microfluidic chamber designs: uniform and binary. Both chambers contain an identical assay area (red dashed-line boxes) where locomotion was analyzed, while the surrounding exploration zones differ in PDMS pillar arrangement. (**B–E**) Quantification of locomotion in the assay area after 1 hr of exploration. Data from binary chambers (*n* = 21) are shown in orange and from uniform (*n* = 21) chambers in blue. Shown are (**B**) locomotion speed, (**C**) percentage of time spent idle, (**D**) reversal frequency, and (**E**) turning frequency. Statistical significance was determined using an unpaired Student's *t*-test (***p < 0.001; *p < 0.05). Mean ± SEM; each point represents the mean behavioral value of all worms within one chamber. (**F**) Percent change in each locomotion metric (speed, idle time, reversal frequency, and turning frequency), calculated from panels B–E and normalized to the mean value in binary chambers. Mean ± SEM; each point represents the mean behavioral value of all worms within one chamber.

The online version of this article includes the following figure supplement(s) for figure 1:

**Figure supplement 1.** Prior exploration of distinct physical settings modulates locomotion speed of worms in identical environments.

context-dependent modulation requires distinct cGMP signaling components. Second, while the microvilli-rich sensory endings of AFD are essential for thermosensation, they are dispensable for context-dependent modulation. Third, this modulation depends on mechanosensory neurons and on electrical synapses between AFD and the AIB interneuron, which connects to mechanosensory circuits. Finally, disruption of *inx-7* and *inx-10* genes, encoding gap junction protein innexins in AFD and AIB, impairs context-dependent locomotion modulation. Remarkably, restoring AFD–AIB connectivity via engineered Cx36 electrical synapses rescues this function. Collectively, our findings uncover an unconventional mechanism by which AFD mediates behavior modulation based on tactile experience.

## Results

### *C. elegans* display distinct locomotion in identical environments after exploring different physical structures

To test whether prior physical experience influences *C. elegans* behavior, we allowed worms to explore microfluidic chambers with distinct pillar designs before recording their locomotion in identical assay zones (*Figure 1A*). Because the environment within the assay zone is the same across chambers, behavioral differences observed there are likely to reflect an influence of prior tactile experience. To implement this assay, we designed two microfluidic chambers that differ only in the physical structure of the exploration zone: a uniform chamber and a binary chamber (*Figure 1A*). Before reaching the assay zone, worms explore surrounding areas with pillar designs that differ in size and spacing

(exploration zone, *Figure 1A*). This design allowed us to assess how exposure to different physical contexts impacts locomotion in an otherwise identical setting.

Worms were allowed to explore the chambers for 60 min before behavior was recorded in the assay zones. We found that worms in the uniform chamber assay zone exhibited locomotor behavior distinct from those in the binary chamber assay zone (*Figure 1B–F*). Specifically, worms in the uniform chamber assay zone moved approximately ~40% faster (*Figure 1B, F*), spent ~20% less time idle (*Figure 1C, F*), reversed direction ~20% less frequently (*Figure 1D, F*), and turned ~30% more often (*Figure 1E, F*) than worms in the binary chamber assay zone. Among the parameters measured, locomotion speed showed the most robust and reproducible modulation, establishing a stable behavioral metric for dissecting the underlying circuit and molecular mechanisms. These findings demonstrate that locomotion is shaped by past sensory experience rather than solely by immediate environmental cues.

Because worms in the binary chamber are exposed to both pillar types and remain free to move between exploration and assay zones, the behavioral differences described above could reflect exposure to a more complex physical environment rather than prior experience alone. To directly test whether locomotion is modulated by prior physical experience independently of continued access to the exploration zone, we designed microfluidic chambers in which the assay zone could be separated from the exploration zone by a removable barrier (*Figure 1—figure supplement 1A*). In these chambers, worms were initially allowed to explore the entire device, including exploration zones that either matched or differed from the assay zone. A barrier was then inserted to prevent worms in the assay zone from re-entering the exploration zones.

Under these conditions, locomotion immediately after barrier insertion was higher in worms that had previously explored physical settings matching the assay zone (205 ± 8 μm/s) than in worms that had explored non-matching settings (151 ± 7 μm/s; p = 0.006; *Figure 1—figure supplement 1B*). This difference persisted when worms were recorded 40 min after barrier insertion, with animals in matching chambers retaining their higher locomotion rates (218 ± 11 μm/s) compared to those in non-matching chambers (185 ± 8 μm/s; p = 0.02; *Figure 1—figure supplement 1B*). These findings demonstrate that prior exploration of distinct physical environments can modulate locomotion even when worms are prevented from returning to those environments, supporting a role for prior physical experience independent of ongoing sensory input.

## GC gene *gcy-18* is required for context-dependent locomotion modulation in chambers

To identify mechanisms underlying context-dependent locomotion adjustments, we employed a previously reported preference assay that uses microfluidic arenas to assess the ability of worms to select a preferred space (*Han et al., 2017*). Because worms must adjust their locomotion based on their surroundings to remain within a preferred area, we hypothesized that mutations abolishing this preference would reveal key mechanisms underlying context-dependent adjustments in the uniform and binary chambers. Accordingly, we screened for mutant worms that failed to choose their preferred area, aiming to identify the genetic mechanisms mediating locomotion adjustments in response to different physical structures.

Given the significant role of cGMP in neuronal plasticity and behavior modulation, we performed a small-scale genetic screen using loss-of-function mutants for 24 genes encoding GCs, the enzymes responsible for cGMP synthesis. Using the preference assay, we identified two putative null mutants, *gcy-18(gk423024)* and *gcy-12(gk142661)*, that displayed significant defects in spatial preference (*Figure 2—figure supplement 1A*). We then compared the locomotion of *gcy-18* and *gcy-12* mutants in both uniform and binary chambers. *gcy-12* mutant worms exhibited normal speed adjustment compared to wild type (Δspeed: *gcy-12*: 24 ± 5% vs. N2: 35 ± 7%, p = 0.27, *Figure 2—figure supplement 1B*), indicating that *gcy-12* is not required for locomotion modulation. In contrast, *gcy-18(gk423024)* mutants failed to execute context-dependent speed adjustment (Δspeed *gcy-18*: 3 ± 5% vs. N2: 35 ± 7%; p = 0.0013; *Figure 2A*). A second allele, *gcy-18(nj38)*, similarly showed significantly reduced Δspeed (11 ± 2%; p = 0.011 compared to wild type; *Figure 2A*). Together, these results indicate that *gcy-18* is required for context-dependent behavioral adjustment.

To confirm that the defects observed in *gcy-18* mutants were not due to background artifacts, we introduced a single-copy *gcy-18* transgene driven by the *gcy-18* promoter into *gcy-18(gk423024)*

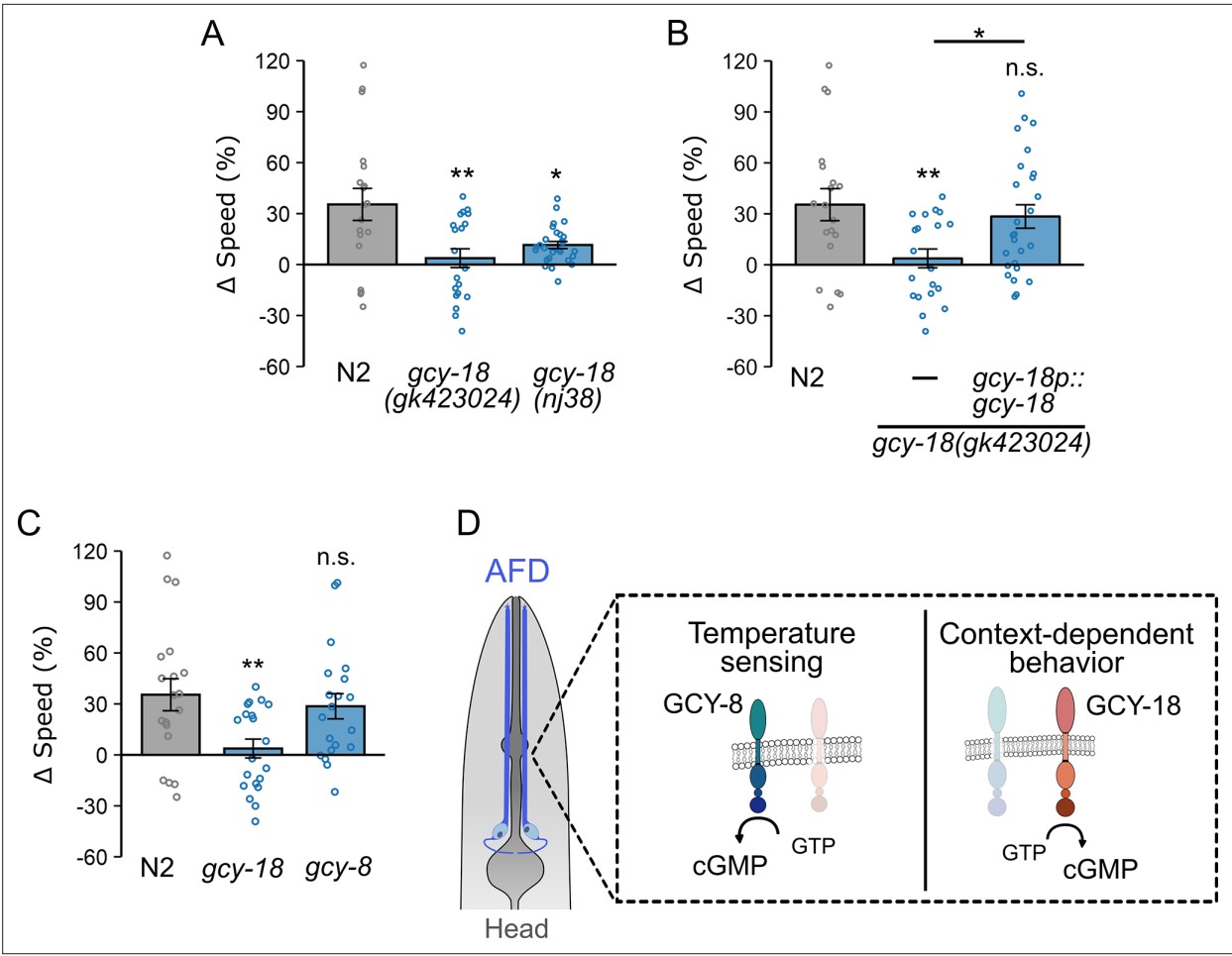

**Figure 2.** guanylyl cyclase gene *gcy-18* functions in AFD to mediate context-dependent locomotion modulation. (**A**) Difference in locomotion speed (Δspeed) between uniform and binary chambers of wild-type N2 (uniform: *n* = 19; binary: *n* = 20), *gcy-18(gk423024)* mutant (uniform: *n* = 20; binary: *n* = 28), and *gcy-18(nj38) mutant* (uniform: *n* = 27; binary: *n* = 27). ΔSpeed was calculated as the percent change in locomotion speed in uniform chambers, relative to the mean speed measured in binary chambers (see Methods). Mutant worms lacking *gcy-18* failed to demonstrate context-dependent locomotion adjustments in locomotion speed, as indicated by a low Δspeed value. (**B**) Δspeed of wild-type, *gcy-18* mutants, and *gcy-18* mutant worms carrying a single-copy transgene expressing *gcy-18* under the control of the *gcy-18* promoter. Expression of *gcy-18p::gcy-18* restored Δspeed (uniform: *n* = 26; binary: *n* = 24). (**C**) Locomotion speed adjustment of N2, *gcy-18* mutants, and *gcy-8* mutant worms (uniform: *n* = 20; binary: *n* = 18). Disruption of *gcy-8* does not impair context-dependent locomotion modulation. (**D**) Schematic illustrating the distinct roles of guanylyl cyclase genes in AFD sensory neurons: *gcy-8* is required for thermosensation but not for context-dependent locomotion modulation, whereas *gcy-18* is essential for context-dependent locomotion modulation and plays only a modest role in thermosensation. Data are presented as mean ± SEM. Each data point represents the mean behavior of worms within a single chamber. Asterisks above bars indicate statistical significance compared to wild type, whereas asterisks above horizontal black lines indicate statistical significance between mutant strains. Statistical significance was determined using one-way ANOVA followed by a Tukey–Kramer post hoc test (n.s., p > 0.05; *p < 0.05; **p < 0.01).

The online version of this article includes the following figure supplement(s) for figure 2:

**Figure supplement 1.** Guanylyl cyclase genes *gcy-18* and *gcy-12* are required for spatial preference, but only *gcy-18* supports context-dependent locomotion modulation.

mutants. Expression of this transgene restored locomotion adjustments to near wild-type levels (Δspeed: *gcy-18p::gcy-18*: 28 ± 6% vs. N2: 35 ± 7%, p = 0.8, *Figure 2B*), confirming that loss of *gcy-18* underlies the observed behavioral defects. The identification of *gcy-18* was particularly intriguing, as its expression is restricted to a single pair of neurons, the thermosensory AFDs. Given that our assay chambers lacked thermal gradients, these findings suggest a previously unrecognized role for *gcy-18* and AFD neurons in mediating locomotion adjustments based on prior exploration of physical environments.

To further investigate the role of AFD-specific GCs in context-dependent locomotion modulation, we examined *gcy-8*, which is essential for generating thermally evoked Ca²⁺ responses in AFD (*Wang et al., 2013*; *Takeishi et al., 2016*). In thermal responses, disrupting *gcy-8* alone is sufficient to impair AFD function, whereas disruption of *gcy-18* alone does not (*Wang et al., 2013*; *Inada et al., 2006*), indicating the critical role of GCY-8 in AFD thermosensation. If *gcy-18* influenced behavioral adjustment in microfluidic chambers solely through its role in thermosensation, we would expect that a loss-of-function mutation in *gcy-8* would have a similar or even more disruptive impact on locomotion modulation. However, *gcy-8* mutant worms exhibit little change in their ability to adjust locomotion speed, with Δspeed values similar to that observed in wild-type worms (Δspeed: *gcy-8*: 29 ± 7% vs. N2: 35 ± 7%; p = 0.96; *Figure 2C*). These results indicate that while *gcy-8* is critical for thermosensation, its mutation does not affect context-dependent locomotion modulation, suggesting that independent cGMP signaling mechanisms govern AFD neuronal activity in thermosensation and context-dependent behavior in microfluidic chambers (*Figure 2D*).

## Distinct CNG channels mediate context-dependent locomotion modulation and thermosensation

The role of GC in cGMP synthesis prompted us to investigate whether CNG channels are involved in context-dependent locomotion modulation. CNG channels, composed of α and β subunits, open upon binding cyclic nucleotides (e.g., cGMP and cAMP), allowing Ca²⁺ influx (*Komatsu et al., 1999*; *Finn et al., 1998*; *Kaupp and Seifert, 2002*; *Xue et al., 2022*) to control signal transduction and neural plasticity across species (*Harris et al., 2014*; *He et al., 2016*; *Herrero et al., 2020*; *Feketa et al., 2020*; *Dolzer et al., 2021*; *Kelliher et al., 2003*; *Michalakis et al., 2011*; *Haverkamp et al., 2006*; *Figure 3A*). In *C. elegans*, mutations in either the *tax-4* or *tax-2* genes, which encode the α and β subunits of CNG channels, respectively, impair thermotaxis behavior (*Mori and Ohshima, 1995*; *Komatsu et al., 1996*) and eliminate AFD responses to temperature changes (*Ramot et al., 2008*; *Kimura et al., 2004*). To determine their role in context-dependent locomotion modulation, we examined *tax-4(p678)* and *tax-2(p694)* mutant worms using our microfluidic chamber assay.

We found that *tax-2* is essential for context-dependent locomotion modulation, as *tax-2* mutants did not exhibit differences in speed between chamber types (*Figure 3B, C*). In contrast, *tax-4* mutants displayed an enhanced locomotion adjustment compared to wild-type worms (Δspeed: *tax-4*: 68 ± 6% vs. N2: 41 ± 6%; p = 0.002; *Figure 3C*), a striking difference from the lack of speed changes seen in *tax-2* mutants (Δspeed: *tax-2*: 2 ± 2%; p < 0.0001 compared to wild type). We examined the locomotion speed of mutant worms in the binary chambers, which we refer to as the basal speed because wild-type worms consistently move slowest in this environment. We found *tax-4* mutants showed significantly higher basal speed than the wild type (*tax-4:* 193 ± 9 μm/s vs. N2: 104 ± 4 μm/s, p < 0.0001; *Figure 3B*), indicating that *tax-4* mutants can still adjust locomotion speed based on prior experience even when basal locomotion is accelerated.

To further examine how basal locomotion influences context-dependent modulation, we took advantage of the fact that worm locomotion rates are sensitive to environmental temperature (*Parida et al., 2014*; *Altanbadralt et al., 2015*). We examined the relationship between temperature-induced changes in basal locomotion and locomotion modulation based on prior experience of physical surroundings. Wild-type worms cultivated at either 23 or 17°C and then transferred to 20°C exhibited significant changes in basal locomotion speed. Despite changes in basal speed, the extent of locomotion modulation in the chambers was largely unaffected (*Figure 3—figure supplement 1A*). Specifically, locomotion speed was positively correlated with temperature shifts in both chambers (binary chambers: r = 0.52, p < 0.0001; uniform chambers: r = 0.48, p < 0.0001; *Figure 3—figure supplement 1B*), whereas no significant correlation was observed between temperature and Δspeed (r = –0.03, p = 0.22; *Figure 3—figure supplement 1C*), supporting the notion that basal locomotion speed and context-dependent modulation are regulated independently.

Together, these observations indicate that TAX-2 is required for integrating prior experience into locomotion modulation, whereas TAX-4 primarily regulates overall locomotion level. Although both pathways engage cGMP signaling, their distinct phenotypes suggest separate regulatory mechanisms for basal locomotion and context-dependent adjustments.

Given that four other genes (*cng-1*, *cng-2*, *cng-3*, and *cng-4*) are predicted to encode CNG channels in *C. elegans*, we investigated their role in locomotion modulation in microfluidic chambers

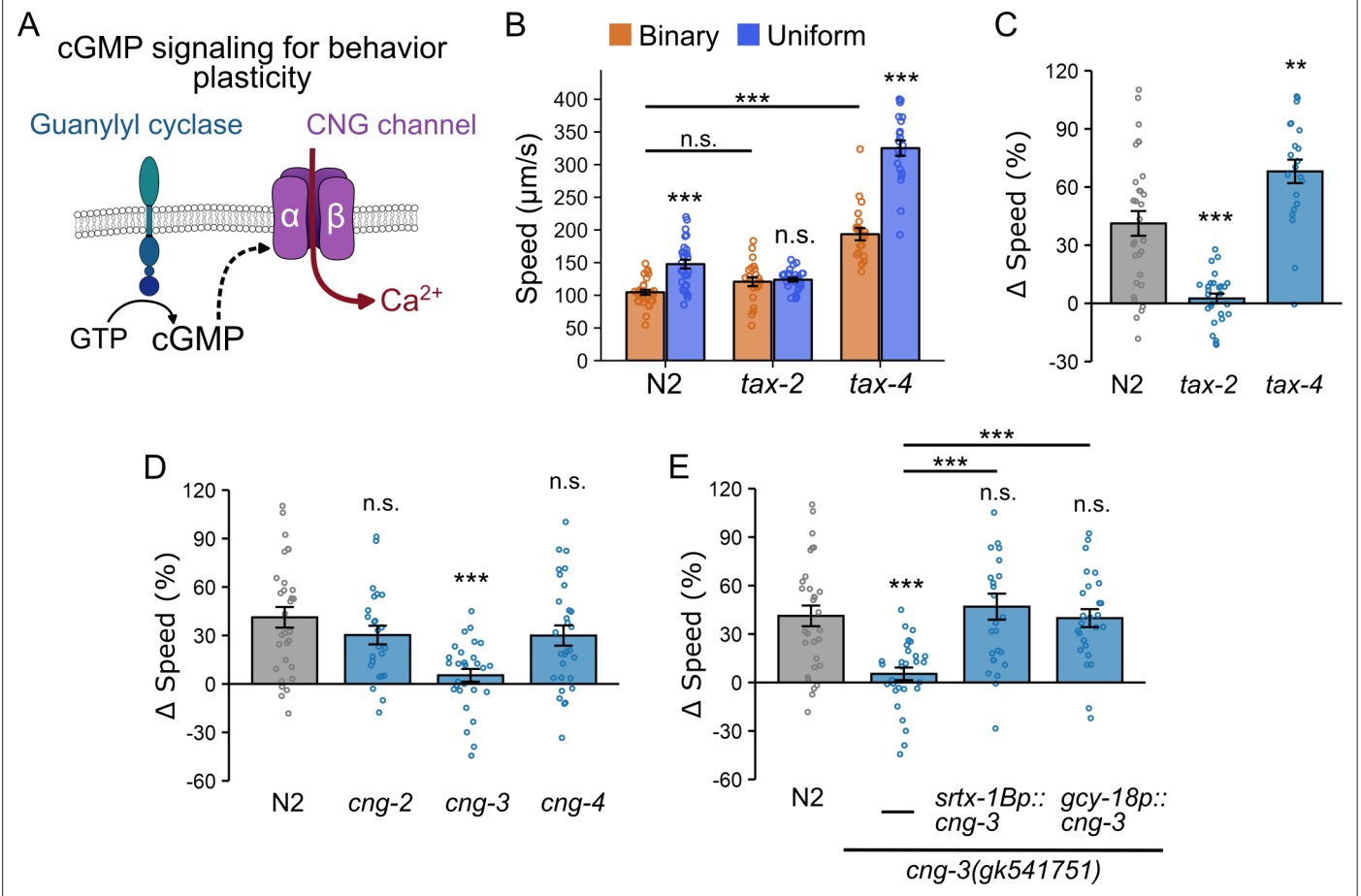

**Figure 3.** Cyclic nucleotide-gated (CNG) channel subunits TAX-2 and CNG-3 are required for locomotion modulation, but TAX-4 is not. (**A**) Schematic representation of CNG channel function. CNG channels are activated by cyclic nucleotides (such as cGMP) and mediate $Ca^{2+}$ influx, thereby influencing neuronal activity and animal behavior. (**B**) Locomotion speed (μm/s) of N2 (uniform: $n = 30$; binary: $n = 29$), $tax-2$ (uniform: $n = 27$; binary: $n = 22$), and $tax-4$ (uniform: $n = 21$; binary: $n = 21$) mutant worms in binary (orange) and uniform (blue) chambers. The $tax-2$ mutants exhibit identical locomotion rates in both chamber types, whereas worms lacking $tax-4$ display accelerated basal locomotion while still preserving context-dependent locomotion modulation. Asterisks above blue bars indicate statistically significant differences in locomotion speed between uniform and binary chambers (unpaired Student's $t$-test). Asterisks above horizontal black lines indicate statistically significant differences in basal speed, defined as speed of worms in the binary chamber, between N2 and $tax-4$ mutants (one-way ANOVA followed by a Tukey–Kramer post hoc test; n.s., $p > 0.05$; ***$p < 0.001$). (**C**) Speed differences (Δspeed) for N2, $tax-2$ mutant, and $tax-4$ mutant worms. The $tax-2$ mutants failed to modulate locomotion rates in a context-dependent manner, whereas $tax-4$ mutations enhanced modulation. (**D**) Assessment of $cng-2$ (uniform: $n = 24$; binary: $n = 22$), $cng-3$ (uniform: $n = 29$; binary: $n = 28$), and $cng-4$ (uniform: $n = 24$; binary: $n = 29$) roles in locomotion adjustments. The $cng-3$ mutation abolishes locomotion modulation. (**E**) Single-copy transgenes expressing $cng-3$ cDNA under two AFD-specific promoters, $srtx-1bp$ (uniform: $n = 26$; binary: $n = 22$) and $gcy-18p$ (uniform: $n = 26$; binary: $n = 25$), restore context-dependent locomotion adjustments. Data are presented as mean ± SEM. Each data point represents the mean behavior of worms within a single chamber. Asterisks denote statistical significance versus wild-type (above bars) or between mutant strains (above horizontal black lines). Δspeed comparisons across strains (panels C–E) were determined using one-way ANOVA followed by Tukey–Kramer post hoc tests (*$p < 0.05$; **$p < 0.01$; ***$p < 0.001$).

The online version of this article includes the following figure supplement(s) for figure 3:

**Figure supplement 1.** Temperature shifts alter absolute locomotion speed but do not affect context-dependent locomotion modulation.

(*Wojtyniak et al., 2013*). Due to abnormal development in $cng-1$ mutants, we were unable to assess their locomotion; however, we were able to evaluate mutants of the other three genes. We found that $cng-3(gk541751)$ mutant worms showed significantly decreased speed modulation compared to wild type (Δspeed: $cng-3$: 5 ± 4%, $p = 0.0002$; *Figure 3D*), while $cng-2(tm4267)$ and $cng-4(gk195496)$ mutants exhibited normal modulation comparable to wild-type worms ($p = 0.54$ and $p = 4.47$, respectively; *Figure 3D*). The identification of CNG-3 was particularly intriguing, as recent studies suggest that it mediates changes in AFD neuronal activity based on prior temperature experiences (*Hill and*

*Sengupta, 2023*; *Cho et al., 2004*), indicating that CNG-3 contributes to neural plasticity rather than direct stimulus detection. Our results further support this view by showing that CNG-3 plays a broader role in locomotion modulation in environments lacking thermal gradients, which suggests its involvement in neuronal plasticity beyond the thermosensory modality.

Next, we determined whether AFD is the site where CNG-3 functions in context-dependent locomotion modulation. To do so, we introduced single-copy transgenes expressing *cng-3* cDNA under the control of two AFD-specific promoters, *srtx-1Bp* and *gcy-18p*, into *cng-3(gk541751)* mutant worms. Regardless of the promoter used, AFD-specific expression of *cng-3* restored the locomotion speed modulation to levels comparable to wild-type (Δspeed: *srx-1Bp::cng-3*: 47 ± 8%; p < 0.0001 compared to *cng-3* mutant, p = 0.91 compared to wild type; and *gcy-18p::cng-3*: 40 ± 6%, p = 0.0007 compared to *cng-3* mutant, p = 0.99 compared to wild type; *Figure 3E*). These results indicate that AFD neurons are the site of action for CNG-3 in mediating context-dependent locomotion modulation. Together, our findings reveal a role for AFD beyond thermosensation and show that components of a cGMP signaling pathway enable AFD to modulate locomotion in an experience-dependent manner.

## AFD sensory microvilli are dispensable for locomotion modulation, but AFD neurons are required

The finding that GCY-18 and CNG-3 in AFD neurons are required for context-dependent locomotion prompted us to test whether the AFD thermosensory apparatus is also necessary for this function. This apparatus consists of sensory microvilli at the tip of the AFD dendrite (*Figure 4A*) and is essential for temperature sensing and thermotaxis (*Yoshida et al., 2016*; *Satterlee et al., 2001*). For example, mutations in the *kcc-3* gene, which encodes a $K^+/Cl^-$ cotransporter in the amphid sheath glia, disrupt microvilli formation and significantly impair thermotaxis toward the cultivation temperature (*Singhvi et al., 2016*).

Despite this disruption, locomotion modulation based on experience of physical settings remained intact in *kcc-3(ok228)* mutant worms compared to wild type (Δspeed: *kcc-3*: 33 ± 4% vs. N2: 42 ± 7%, p = 0.28; *Figure 4B, C*). *kcc-3* mutants also displayed elevated basal locomotion rates (basal speed: *kcc-3*: 174 ± 8 μm/s vs. N2: 115 ± 7 μm/s, p < 0.0001; *Figure 4B*), similar to what we observed in *tax-4* mutants. This supports our earlier finding that overall locomotion rate does not necessarily correlate with context-dependent modulation.

To further test whether AFD sensory endings are required for context-dependent locomotion modulation, we examined *ttx-1(767)* mutant worms, which completely lack AFD microvilli and exhibit more severe thermotaxis defects than *kcc-3* mutants (*Satterlee et al., 2001*). Despite the loss of sensory microvilli, *ttx-1* mutants retained the ability to modulate locomotion in microfluidic chambers. The extent of modulation was modestly reduced and more variable compared to wild-type animals (Δspeed: *ttx-1*: 29 ± 10% vs. N2: 42 ± 7%, p = 0.29, *Figure 4D*). Because *ttx-1* mutations disrupt the expression of multiple AFD genes, including components of the cGMP signaling pathway (*Kagoshima and Kohara, 2015*), this increased variability may reflect broader perturbations in AFD function. Nonetheless, these results indicate that AFD sensory microvilli are not strictly required for context-dependent locomotion modulation.

Given that *gcy-18* is required for context-dependent locomotion modulation and that GCY-18 localizes to the distal dendrite of AFD, we next examined how disruption of sensory microvilli affects its localization in AFD. We used a split-GFP strategy to visualize endogenous GCY-18 (*He et al., 2019*). A tandem array of seven GFP11 β-strands (GFP11x7) was inserted at the C-terminus of GCY-18 using CRISPR–Cas9. When complemented with GFP1-10, GCY-18::GFP11x7 fluorescence was strongly enriched at the AFD sensory microvilli near the nose (*Figure 4—figure supplement 1A–A''*), consistent with previous reports (*Takeishi et al., 2016*; *Harris et al., 2023*; *Nguyen et al., 2014*). In addition, weaker but reproducible GCY-18 signal was detected near the AFD soma and axon (*Figure 4—figure supplement 1A'''*). Importantly, in *kcc-3*, which exhibit disrupted sensory microvilli, and *ttx-1* mutants, which lack sensory microvilli, GCY-18 remained localized to the distal dendrite and was still detectable near the soma and axon (*Figure 4—figure supplement 1B–B''' and C–C'''*). Although these experiments do not identify the precise subcellular site at which GCY-18 acts, they show that disruption or loss of sensory microvilli does not substantially alter GCY-18 localization within AFD.

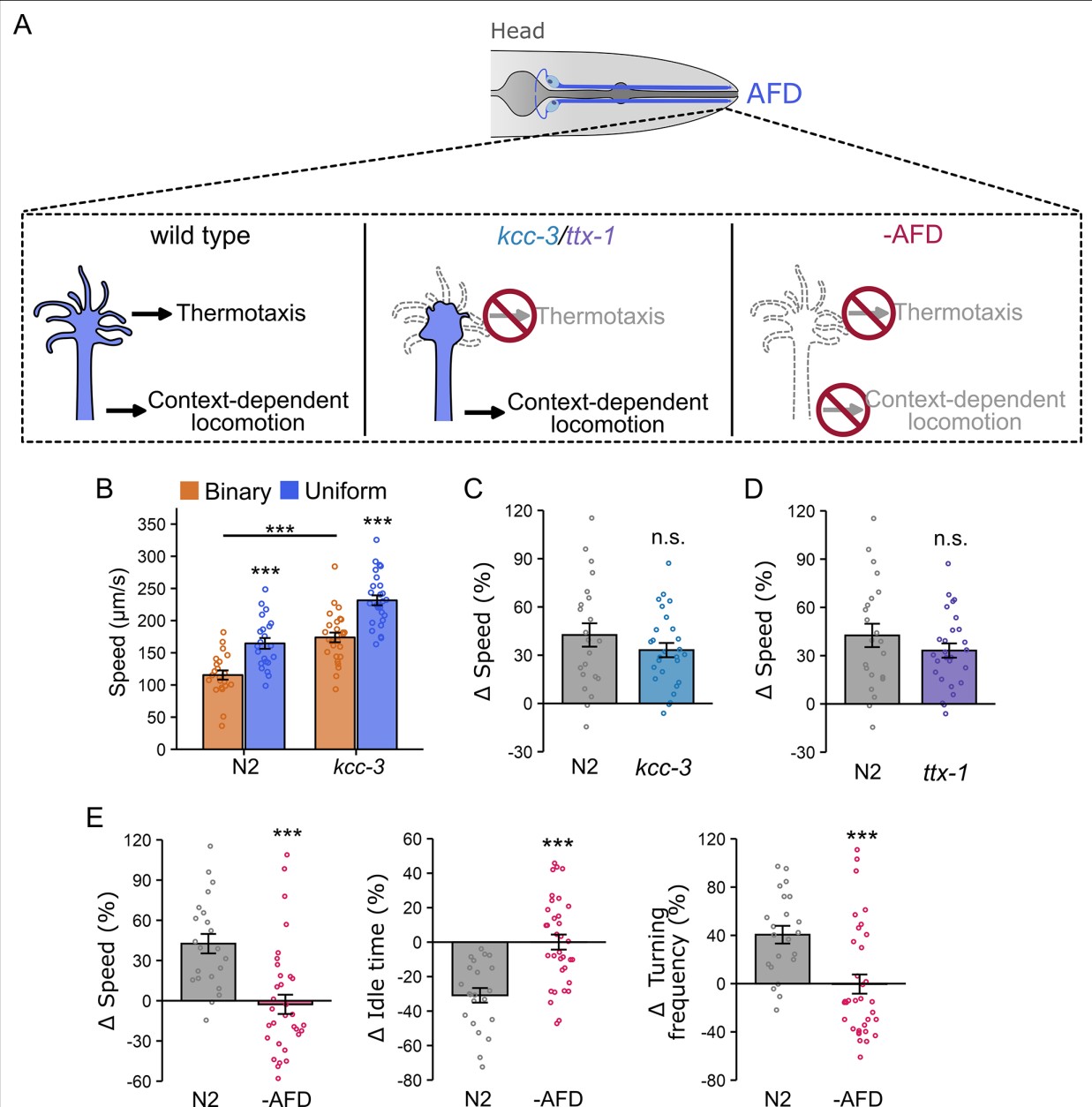

**Figure 4.** AFD, but not its sensory endings, is required for context-dependent locomotion modulation. (**A**) Schematic representation of AFD function in temperature sensing and locomotion modulation. AFD sensory endings are essential for thermosensation but dispensable for context-dependent locomotion modulation. (**B**) Locomotion speed (μm/s) of wild-type N2 (uniform: *n* = 22; binary: *n* = 22) and *kcc-3* (uniform: *n* = 27; binary: *n* = 27) mutant worms in the binary (orange) and uniform (blue) chambers. The *kcc-3* mutants preserve context-dependent locomotion modulation while exhibiting increased basal locomotion rates. Asterisks above blue bars indicate significant differences between uniform and binary chambers (unpaired Student's *t*-test). Basal speed differences in the binary chambers were determined using one-way ANOVA followed by Tukey–Kramer post hoc tests (***$p < 0.001$). (**C**) Δspeed of N2 and *kcc-3* mutant worms, and (**D**) Δspeed of N2 (uniform: *n* = 22; binary: *n* = 22) and *ttx-1* (uniform: *n* = 27; binary: *n* = 27) mutant worms. Context-dependent locomotion modulation remains in *kcc-3* and *ttx-1* mutant worms, although they abolish the AFD thermosensory function. (**E**) Ablation of AFD eliminates the context-dependent modulation of speed, idle time, and turning frequency (uniform: *n* = 34; binary: *n* = 36). Data are presented as mean ± SEM. Each data point represents the mean behavior of worms within a single chamber. Asterisks indicate statistical significance compared to wild type. Δspeed across strains (panels C–E) was analyzed by one-way ANOVA followed by Tukey–Kramer post hoc tests (n.s., $p > 0.05$; ***$p < 0.001$).

The online version of this article includes the following figure supplement(s) for figure 4:

**Figure supplement 1.** GCY-18 primarily localizes to AFD dendritic endings and remains detectable in *kcc-3* and *ttx-1* mutants.

**Figure supplement 2.** Neither *gcy-18* mutation nor AFD ablation causes gross developmental or motor defects.

We next asked whether AFD neurons themselves are required for this behavior. Using a published strain expressing caspase under the AFD-specific *gcy-8* promoter (***Glauser et al., 2011***), we found that AFD-ablated worms completely lost the ability to adjust locomotion in microfluidic chambers. These animals showed no detectable differences between uniform and binary chambers in locomotion speed, idle time, or turning frequency (***Figure 4E***). Together, these results indicate that while AFD sensory microvilli are dispensable for context-dependent locomotion modulation, the AFD neurons themselves are essential.

Building on our finding that locomotion modulation can be driven by prior physical experience even after worms are prevented from re-entering the exploration zones, we next tested whether AFD is required for this modulation using chambers in which the exploration and assay zones were separated by a removable barrier (***Figure 1—figure supplement 1A***). Under these conditions, locomotion modulation was significantly reduced in AFD-ablated worms (Δspeed: −AFD = 1 ± 6% vs. N2 = 23 ± 7%; p = 0.036; ***Figure 4—figure supplement 2A***). Similarly, *gcy-18* mutants showed defective locomotion modulation (Δspeed: *gcy-18* = −1 ± 8% vs. N2 = 23 ± 7%; p = 0.034; ***Figure 4—figure supplement 2A***). These results indicate that AFD and *gcy-18* are required to generate locomotion modulation in response to recent physical experience.

Finally, to determine whether the modulation defects observed in *gcy-18* mutants and AFD-ablated worms could be attributed to developmental abnormalities or gross motor impairments, we measured locomotion speed and body length on standard nematode growth medium (NGM) plates. Both day-1 adult AFD-ablated worms (speed: 281 ± 10 μm/s; p = 0.33; body length: 1.12 ± 0.01 mm; p = 0.76) and *gcy-18* mutants (speed: 291 ± 13 μm/s; p = 0.22; body length: 1.15 ± 0.02 mm; p = 0.86) showed locomotion speeds and body lengths comparable to wild-type controls (speed: 252 ± 30 μm/s; body length: 1.14 ± 0.02 mm; ***Figure 4—figure supplement 2B, C***). These results indicate that the loss of context-dependent locomotion modulation is not due to developmental defects or gross impairments in locomotion.

## Context-dependent locomotion modulation requires the MEC-10 mechanosensory channel subunit

Having established that AFD integrates prior physical experience independent of its thermosensory apparatus, we next asked whether direct mechanosensory input is required for context-dependent locomotion modulation. The *mec-10* gene encodes an amiloride-sensitive Na$^+$ channel protein required for sensing body touch. Mutations in *mec-10* disrupt mechanotransduction channels in the worms, impairing touch sensitivity (***Arnadóttir et al., 2011***; ***Chatzigeorgiou et al., 2010***). We found that *mec-10(tm1552)* mutants failed to display locomotion adjustment compared to wild-type worms (Δspeed: *mec-10*: 5 ± 4% vs. N2: 33 ± 6%, p = 0.0003; ***Figure 5A***). These results show that *mec-10* is crucial for context-dependent locomotion modulation, supporting the hypothesis that locomotion is modulated depending on the worm's tactile experience.

The *mec-10* gene is expressed in several mechanosensory neurons, including the six touch receptor neurons (TRNs) and the polymodal nociceptors FLP and PVD (***Arnadóttir et al., 2011***; ***Huang and Chalfie, 1994***). To determine which neurons are required for tactile-dependent locomotion modulation, we expressed *mec-10* cDNA under cell-specific promoters: *mec-18p* (TRNs) (***Zhang et al., 2004***), *egl-44p* (FLP) (***Wu et al., 2001***), or *mec-10p* (TRNs, FLP, and PVD) (***Huang and Chalfie, 1994***). Expression in either FLP or TRNs alone did not restore modulation, as worms carrying *egl-44p::mec-10* (Δspeed: −11 ± 4%) or *mec-18p::mec-10* (Δspeed: −13 ± 4%) transgenes showed significantly reduced Δspeed compared to wild type (Δspeed: N2: 33 ± 6%; p < 0.0001 for both; ***Figure 5A***). By contrast, *mec-10* co-expression in both FLP and TRNs (Δspeed: 16 ± 4%), or expression from the *mec-10* promoter (Δspeed: 23 ± 4%), restored Δspeed to wild-type levels (p = 0.20 and p = 0.57, respectively; ***Figure 5A***). These findings indicate that *mec-10* expression across multiple mechanosensory neuron types is required for context-dependent locomotion modulation. It is also worth noting that, while both tactile-dependent locomotion modulation and previously reported spatial preference require FLP, only the former depends on TRNs. Together, these findings suggest that distinct subsets of mechanosensory neurons differentially contribute to behaviors shaped by tactile experience.

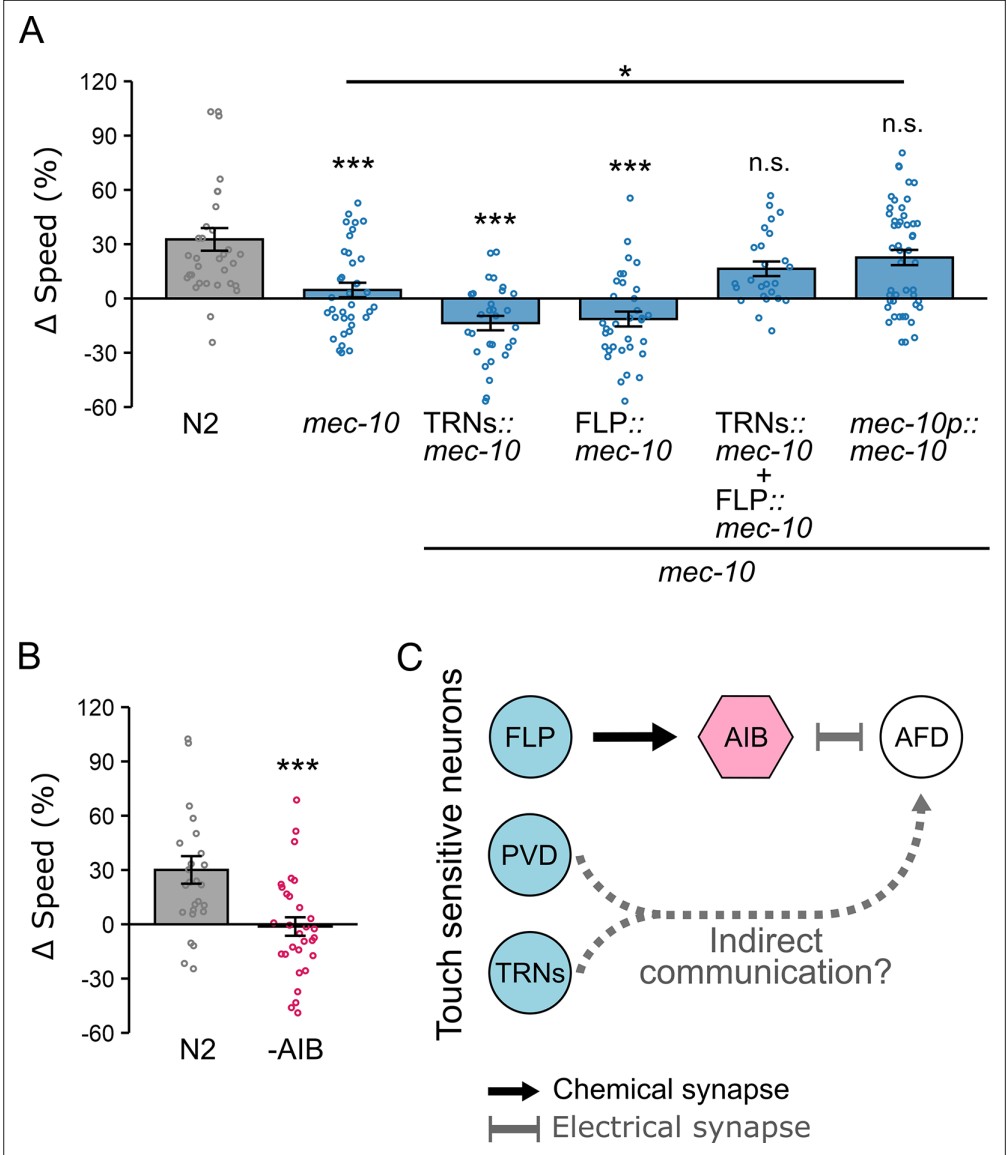

**Figure 5.** Context-dependent locomotion adjustments require the mechanosensory channel subunit MEC-10 and the interneuron AIB. (**A**) Δspeed of wild-type (uniform: *n* = 26; binary: *n* = 22), *mec-10* mutants (uniform: *n* = 30; binary: *n* = 21), and *mec-10* mutants expressing *mec-10* cDNA under cell-specific promoters: *mec-18p* (TRNs; uniform: *n* = 29; binary: *n* = 25), *egl-44p* (FLP; uniform: *n* = 27; binary: *n* = 34), or *mec-10p* (TRNs, FLP, and PVD; uniform: *n* = 32; binary: *n* = 26). (**B**) Δspeed of wild-type and AIB-ablated worms (uniform: *n* = 30; binary: *n* = 30). AIB ablation abolishes speed modulation. Data are shown as mean ± SEM; each data point represents the average speed of worms in a single chamber. Asterisks indicate statistical significance compared to wild type (one-way ANOVA followed by Tukey–Kramer post hoc tests; ***p < 0.001). (**C**) Schematic illustrating a circuit framework connecting *mec-10* expressing neurons, AIB, and AFD in tactile-dependent modulation of locomotion.

## AIB interneurons are crucial for tactile-dependent behavioral plasticity

Touch-sensitive neurons that express *mec-10*, including TRNs, FLP, and PVD, do not form direct synapses with AFD, suggesting that tactile information is relayed through intermediary neurons. Because the interneuron AIB receives synaptic input from FLP and forms electrical synapses with AFD, we hypothesized that AIB could serve as a conduit for mechanosensory signals to reach AFD. To test whether AIB is required for tactile-dependent modulation, we examined locomotion in worms with genetically ablated AIB neurons using *npr-9p::caspase* expression (*Kunitomo et al., 2013*). AIB-ablated worms failed to adjust locomotion speed, showing a near-complete loss of modulation

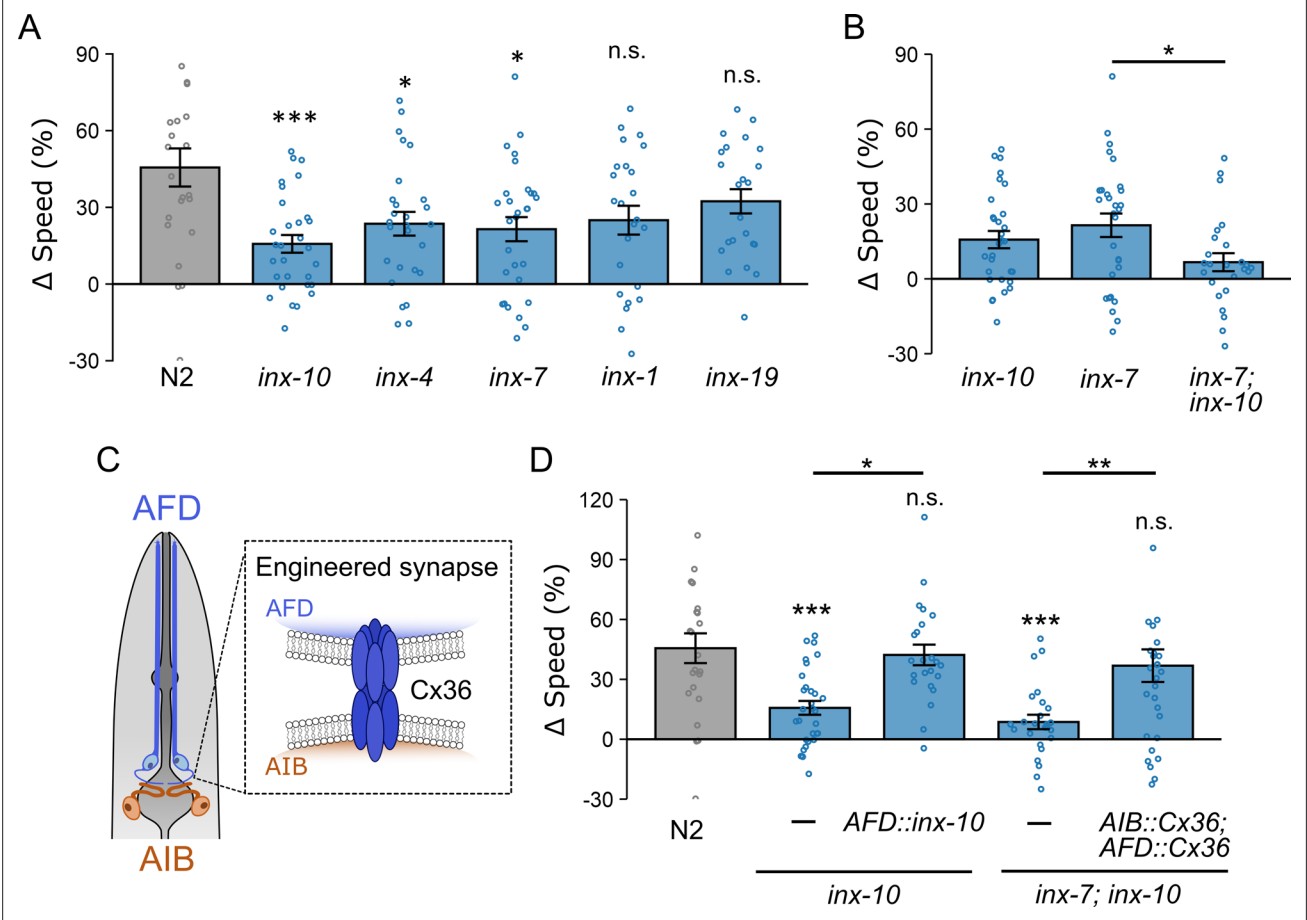

**Figure 6.** Tactile-dependent locomotion modulation is disrupted in mutant worms lacking gap junction genes and is restored by engineered Cx36 electrical synapses linking AFD and AIB. (**A**) Speed differences (Δspeed) for wild-type worms (uniform: *n* = 23; binary: *n* = 25) and *inx-1* (uniform: *n* = 24; binary: *n* = 24), *inx-4* (uniform: *n* = 27; binary: *n* = 26), *inx-7* (uniform: *n* = 29; binary: *n* = 30), *inx-10* (uniform: *n* = 30; binary: *n* = 29), *inx-19* (uniform: *n* = 23; binary: *n* = 24) mutants. (**B**) Speed differences for *inx-7; inx-10* double mutants (uniform: *n* = 25; binary: *n* = 27). (**C**) Schematic illustrating engineered electrical synapses formed by Cx36 when expressed in adjacent neurons. (**D**) AFD-specific expression of *inx-10* cDNA restored tactile-dependent locomotion modulation in *inx-10* mutant worms (*srtx-1bp*; uniform: *n* = 23; binary: *n* = 24). Expression of Cx36 in both AFD (*srtx-1bp*) and AIB (*inx-1p*) similarly restored modulation in *inx-7; inx-10* double mutants (uniform: *n* = 30; binary: *n* = 28). Data are presented as mean ± SEM. Asterisks denote statistical significance versus wild type (above bars) or between indicated genotypes (above horizontal black lines). Each data point represents the mean behavior of worms within a single chamber. Statistical significance was determined using one-way ANOVA followed by Tukey–Kramer post hoc tests (*p < 0.05; **p < 0.01; ***p < 0.001).

(Δspeed: –1 ± 5%) compared to wild type (30 ± 8%, p = 0.001, *Figure 5B*). These results demonstrate that AIB is required for AFD-mediated tactile-dependent locomotion modulation. However, because *mec-10*-expressing TRNs are also required, additional pathways beyond AIB likely contribute to transmitting tactile information to AFD, potentially involving indirect synaptic connections through other interneurons or long-distance signaling via neuropeptides or other modulators (*Figure 5C*).

## Mutations in *innexin* genes disrupt tactile-dependent modulation

Electrical synapses between AIB and AFD neurons in *C. elegans* are formed by gap junction proteins known as innexins (*Güiza et al., 2018*). If electrical coupling between AIB and AFD contributes to tactile-dependent locomotion modulation, then disruption of innexin genes should impair locomotion adjustments in microfluidic chambers. To test this, we examined loss-of-function mutants of several innexin genes. *C. elegans* possesses 25 innexin genes (*Altun et al., 2009*; *Phelan and Starich, 2001*). Single-cell RNA-seq data from the CeNGEN database (*Taylor et al., 2021*) indicate that three genes (*inx-4*, *inx-10*, and *inx-19*) are expressed in AFD, one (*inx-1*) is expressed in AIB, and one (*inx-7*) is expressed in both. We found that *inx-4*, *inx-7*, and *inx-10* mutants exhibited a partial but

significant reduction in locomotion modulation compared to wild-type worms (Δspeed: N2: 45 ± 7%; *inx-4*: 23 ± 5%, p = 0.037; *inx-7*: 21 ± 5%, p = 0.013; *inx-10*: 16 ± 3%, p = 0.0007; *Figure 6A*). Mutations in *inx-1* also reduced Δspeed values, although the effect did not reach statistical significance (Δspeed: *inx-1:* 25 ± 6%, p = 0.076), whereas *inx-19* mutations had little impact on speed modulation (*inx-19:* 32 ± 5%, p = 0.51, vs. wild type; *Figure 6A*). Furthermore, *inx-7; inx-10* double mutants exhibited a more pronounced reduction in Δspeed compared to single mutants (Δspeed: *inx-7; inx-10* double mutant: 6 ± 5%; p = 0.032 compared to *inx-7* single mutant; *Figure 6B*). Together, these findings suggest that multiple innexin genes act redundantly to support tactile-dependent locomotion modulation.

### Targeted electrical coupling between AFD and AIB restores tactile-dependent modulation

We next tested whether restoring *inx-10* specifically in AFD would be sufficient to rescue the behavioral defect. Using the AFD-specific *srtx-1b* promoter, we expressed *inx-10* cDNA in *inx-10* mutant worms. These transgenic animals displayed significantly improved locomotion modulation (Δspeed: 42 ± 5%) compared to non-transgenic *inx-10* mutants (15 ± 4%; p = 0.018; *Figure 6D*), indicating that *inx-10* expression in AFD alone is sufficient to restore function.

To directly test whether restoring the electrical connection between AFD and AIB could rescue modulation in innexin mutants, we expressed the mammalian gap junction protein Cx36 (*Rabinowitch et al., 2021*; *Choi et al., 2020*; *Rabinowitch et al., 2014*; *Rabinowitch and Schafer, 2015*) in both AFD (*srtx-1Bp*) and AIB (*inx-1p*) neurons in *inx-7; inx-10* double mutants (*Figure 6C*). Cx36 expression significantly improved locomotion modulation, restoring Δspeed from 7 ± 4% to 35 ± 8% (p = 0.0035; *Figure 6D*). These results demonstrate that although innexin mutations disrupt connectivity across multiple neurons, restoring a single targeted electrical connection between AFD and AIB is sufficient to rescue tactile-dependent locomotion modulation, identifying the AIB–AFD electrical synapse as a critical circuit element for tactile-dependent behavioral control.

## Discussion

Our study reveals that the thermosensory AFD neuron plays an unexpected role in coupling tactile experiences to locomotion modulation in *C. elegans*. We show that (1) *C. elegans* exhibit distinct locomotion patterns in identical environments, influenced by prior tactile experiences; (2) tactile-dependent behavior modulation requires the AFD neuron but not its thermosensory dendritic endings; (3) thermotaxis and tactile-dependent locomotion modulation rely on distinct sets of cGMP signaling proteins and CNG channel subunits; (4) the AIB interneuron, which connects AFD to mechanosensory circuits, is essential for tactile modulation; and (5) electrical synapses between AFD and AIB are critical for this process. Below, we discuss the implications of these findings.

### A thermosensation-independent role for AFD in tactile-dependent behavior modulation

Our findings highlight the remarkable versatility of sensory neurons in shaping behaviors beyond their primary functions. This supports the emerging view that sensory neurons have multifaceted roles in circuit modulation, extending well beyond canonical sensory activities. Such complexity is encoded at multiple levels within neural circuits. In *C. elegans*, for instance, the ASH neuron drives avoidance responses to various noxious stimuli, including hyperosmolarity, nose touch, and volatile repellents, demonstrating polymodal functionality (*Chang et al., 2006*; *Hart et al., 1995*; *Kaplan and Horvitz, 1993*; *Yoshida et al., 2012*). Similarly, the olfactory receptor neuron AWC exhibits stochastic responses to temperature, introducing variability that enhances the flexibility and reliability of thermotactic behavior (*Biron et al., 2008*; *Kuhara et al., 2008*; *Kano et al., 2023*). In contrast to these examples, the role of AFD in tactile-dependent plasticity does not rely on its sensory endings but is mediated through synaptic connectivity. This reveals an unexpected mechanism by which AFD influences context-dependent behavioral plasticity via network-level interactions rather than direct sensory activity.

## Distinct cGMP signaling pathways in AFD mediate sensation and plasticity

Our data indicate that the CNG channel α subunits TAX-4 and CNG-3 play distinct roles in AFD function. Although TAX-4 is essential for AFD to detect temperature and generate thermally evoked responses (*Ramot et al., 2008*), it is dispensable for tactile-dependent locomotion modulation. By contrast, CNG-3, which is not required for temperature sensing (*Hill and Sengupta, 2023*), is crucial for tactile-dependent locomotion modulation. Consistent with a broader role in sensory plasticity, CNG-3 is required in AWC neurons for short-term odor adaptation (*O'Halloran et al., 2017*) and in AFD for adjusting temperature thresholds based on prior thermal experiences (*Hill and Sengupta, 2023*). Our demonstration that CNG-3 is essential for tactile-dependent locomotion modulation further supports the view that CNG-3 enables context-dependent behavioral plasticity rather than directly mediating stimulus detection.

CNG channels are heteromeric complexes composed of both α and β subunits. Our data, along with previous work, show that TAX-2, the CNG β subunit in *C. elegans*, is required for both tactile-dependent locomotion modulation and thermotransduction (*Ramot et al., 2008*). Interestingly, TAX-4, the canonical α partner of TAX-2, is not required for tactile-dependent modulation. This suggests that TAX-2 may support experience-dependent neural plasticity either on its own or by pairing with alternative α subunits such as CNG-3. This type of functional specialization is reminiscent of photoreceptor neurons, where the composition and ratio of CNG channel subunits determine light sensitivity and enable contrast detection under varying light intensities (*Peng et al., 2004*; *Shuart et al., 2011*; *Zhong et al., 2002*). It is worth noting that the *tax-2(p694)* mutation affects *tax-2* expression in four neurons (AFD, BAG, ASE, and ADE) (*Coburn and Bargmann, 1996*). Moreover, the strong convergence of functional and circuit-level evidence, including the requirement for GCY-18, the rescue of CNG-3 function in AFD, and the role of AIB–AFD coupling, points to AFD as the most likely site of TAX-2 function in tactile-dependent locomotion modulation.

In addition to the diversity of CNG channels, GCs in AFD also exhibit functional specificity. For example, mutations in *gcy-18* alone significantly disrupt tactile-dependent locomotion modulation without substantially affecting temperature sensing (*Wang et al., 2013*; *Inada et al., 2006*), whereas *gcy-8* is essential for generating thermally evoked $Ca^{2+}$ responses (*Wang et al., 2013*; *Takeishi et al., 2016*) yet appears dispensable for tactile-dependent plasticity. These results show that AFD uses distinct sets of GCs to execute dual roles: precise temperature sensing and tactile-dependent behavioral modulation.

## Differential regulation of context-dependent plasticity and basal locomotion rates in AFD

Our results further suggest that the mechanisms controlling basal locomotion and context-dependent behavioral modulation are not only distinct but also operate independently. Temperature fluctuations significantly affect basal locomotion rates, that is, higher temperatures accelerate speed, while lower temperatures slow it down (*Parida et al., 2014*; *Altanbadralt et al., 2015*). However, tactile-dependent locomotion modulation in microfluidic chambers remains robust despite these temperature-induced changes in basal movement. This independence is further supported by our observations in *tax-4* and *kcc-3* mutants, which display elevated basal locomotion rates yet still adjust their locomotion in response to tactile experiences. These findings indicate that the processes governing basal locomotion and context-dependent modulation are controlled by separate, non-interfering mechanisms, even though both pathways engage cGMP signaling.

## Electrical synapses link the thermosensory neuron AFD to tactile-dependent behavioral modulation

AIB–AFD electrical synapses are pivotal for tactile-dependent behavioral plasticity. Disrupting this connection impairs tactile-dependent locomotion modulation, whereas restoring an engineered electrical synapse rescues this behavior. The successful replacement of invertebrate innexins with vertebrate connexins, which perform analogous functions despite lacking sequence homology, indicates that the core function of these synapses lies in their connectivity rather than in their specific protein sequences. This observation suggests that the transfer of signaling molecules such as $Ca^{2+}$ and cGMP

through electrical synapses may facilitate the coordination of neuronal activity across distinct networks (*Yuan et al., 2023*; *Moore et al., 2020*; *Vaughn and Haas, 2022*; *Jin et al., 2020*).

Together, these findings support a model in which AIB functions as a hub neuron that relays mechanosensory input from FLP to AFD to modulate locomotion (*Figure 5C*). However, because electrical synapses are often bidirectional, information flow may also occur in the opposite direction, from AFD to AIB. Determining the directionality and dynamics of this signaling remains an important question for future investigation. A similar circuit logic has been proposed for thermal avoidance behavior, in which AIB integrates inputs from FLP via chemical synapses and couples to AFD via electrical synapses, forming an FLP–AIB–AFD circuit. In that case, the directionality of information flow also remains unresolved (*Liu et al., 2012*).

Finally, the proposed circuit mechanism for tactile-dependent modulation does not exclude additional routes of modulation. These may include extra-synaptic signaling mechanisms, such as neuropeptide release from other touch-sensitive neurons. While the precise molecular and circuit mechanisms remain to be elucidated, our findings establish that AIB–AFD electrical synapses are essential for incorporating tactile context into locomotor control.

### In summary

Our findings reveal an unexpected role for AFD in tactile-dependent behavioral plasticity. They show that specialized molecular mechanisms, including distinct cGMP signaling pathways, diverse GCs, and functional electrical synapse connectivity, enable AFD sensory neurons to perform dual functions: detect stimuli and support tactile-dependent plasticity. A limitation of this study is that the directionality and mode of information flow between AFD and AIB remain unresolved, and defining this relationship will be an important goal for future investigation. Nevertheless, our results establish the AFD–AIB connection as a regulatory hub in tactile-dependent locomotor modulation and expand our understanding of sensory neuron versatility in adaptive behavior.

## Materials and methods

### Nematode strains and growth

*C. elegans* strains were maintained under standard conditions at 20°C on NGM agar plates (*Brenner, 1974*). These plates were seeded with *E. coli* OP50 lawns. Strains used in this study are listed in Key Resources Table and are available from the corresponding author upon request.

### Molecular biology

All DNA constructs were sequence-verified. cDNA sequences encoding *mec-10*, *cng-3*, and *inx-10* were amplified from a cDNA library made with the SuperScript III Reverse Transcriptase kit (Thermo Fisher Scientific, MA) from total RNA extracted from adult N2 worms. A codon-optimized cDNA sequence for the mouse gap junction protein connexin-36 (Cx36) was previously described (*Rabinowitch et al., 2021*). DNA plasmid constructs for *C. elegans* expression were assembled by LR recombination using the MultiSite Gateway Three-Fragment Vector Construction Kit (Thermo Fisher Scientific, MA). Entry clones were constructed as follows: (1) promoter sequences flanked by attL4 and attR1 sites; (2) cDNA sequences flanked by attL1 and attL2 sites; and (3) *sl2::mNeonGreen::rab-3 3'utr* flanked by attR2 and attL3 sites. Destination vectors used for LR recombination included the pDEST R4-R3 Vector II and a custom-built destination vector BJP-C957, which contains a hygromycin resistance cassette for selection, an intestinal fluorescent marker (*vha-6p::mScarlet::tbb-2 3'utr*) for visual screening transgenic worms, and flanking *mos1* sequences to facilitate homologous recombination for Mos1-mediated single-copy insertion (MosSCI). Promoters used for tissue- or cell-specific expression were: *gcy-18p* (2 kb) and *srtx-1Bp* (493 bp) for AFD; *mec-18p* (390 bp) for TRNs; *egl-44p* (5 kb) for FLP; *inx-1p* (1 kb) for AIB; and *mec-10p* (3 kb) for expression in TRNs, FLP, and PVD. Plasmids used to generate transgenic strains in this study are listed in Key Resources Table and are available from the corresponding author upon request.

### Transgenes, germline transformation, and genome editing

Strains carrying extrachromosomal arrays were generated by microinjecting various plasmids together with co-injection markers. The co-injection markers included *unc-112p::gfp* (30 ng/µl), *rpl-28p::neoR*

(20 ng/μl), and *Cbr-unc-119* (20 ng/μl). The blank vector pBluescript was used as filler DNA to bring the final total DNA concentration to 100 ng/μl. A detailed list of the plasmids used to create transgenic worms in this study is provided in Key Resources Table.

Mos1-mediated transgene insertion (*Frøkjær-Jensen et al., 2012*; *Zhang et al., 2022*) was used to generate single-copy transgenic animals. Briefly, worms carrying the *ttTi5605 Mos1* site on chromosome II were injected with a plasmid mixture containing the transgene of interest (40 ng/μl), along with BJP-B908, a plasmid expressing Cas9 and a guide RNA targeting the Mos1 site (60 ng/μl). Selection markers included *rps-0p::hygR* for hygromycin resistance and *vha-9p::mScarlet* for intestinal red fluorescence to enable visual screening.

The *gcy-18::gfp11×7* knock-in strain was generated using a modified CRISPR–Cas9 protocol employing preassembled Cas9 ribonucleoprotein (RNP) complexes, co-injected with a single-stranded DNA repair template to facilitate homologous recombination (*Ghanta et al., 2021*; *Paix et al., 2015*). Cas9 protein, tracrRNA, and crRNA (targeting sequence: TACAACGACTGAAAGAGGAG) were obtained from Integrated DNA Technologies (IDT, NJ). The single-stranded DNA repair template for generating *gcy-18::gfp11×7* was synthesized by Twist Bioscience. Successful editing produced the *pek331* allele, which carries an in-frame GFP11×7 insertion at the C-terminus of GCY-18. The insertion was verified by PCR and DNA sequencing. A worm strain, BJH833, carrying both *pek331* and *muIs253 [eft-3p::sfgfp1-10::unc-54 3'utr],* was used to visualize GCY-18 by complemented split-GFP fluorescence (*He et al., 2019*).

## Fabrication of PDMS microfluidic chambers

Microfluidic chamber designs were created in AutoCAD and fabricated as 100 μm SU8-silicon molds using photolithography (FlowJem, NO, Canada). PDMS devices were produced by pouring degassed PDMS prepolymer (Sylgard 184, Dow Corning, MI) onto the molds, degassing again under vacuum (–25 inHg, 15 min, room temperature), and curing at 80°C for 2 hr. The cured PDMS was peeled from the mold, and inlet/outlet ports were punched using a 2-mm biopsy punch (Acuderm Inc, FL). For the chambers with a removable barrier shown in *Figure 1—figure supplement 1A*, cast PDMS chamber structures were bonded onto a thin semi-cured PDMS sheet to create a permanent, watertight seal. To prepare the sheet, ~30 ml of PDMS was poured into a 6-in. Petri dish, degassed for 15 min, and partially cured at 65°C for 15–20 min, until firm but still tacky. PDMS chamber structures were gently placed onto the sheet with ~2 mm spacing to allow the formation of a barrier, then fully cured at 80°C for 30 min. PDMS devices were bonded to 1 mm glass microscope slides using oxygen plasma treatment (PDC-32G, Harrick Scientific, NY, 1 min).

## Behavior recording and analysis

Videos were captured for 10 s at 7.5 frames per second using a Basler acA2440 camera (Basler, PA) equipped with an AF Micro-NIKKOR 60 mm lens (Nikon, NY). Behavioral data were extracted from the videos using WormLab software (MBF Bioscience, VT) through individual worm tracking. Only those worms that were already within the assay area at the start of the recording were analyzed, and any worms that entered later were excluded. Crawling speed was determined by the total distance traveled by a worm during the 10-s recordings. A turn was defined as a change in crawling direction of 15° or more. Reversals were defined as sudden movement in the opposite direction for at least 5 frames (~0.6 s). Worms were considered idle if their absolute velocity (distance traveled between frames) was below 20 μm/s for more than 10 frames (~1.3 s).

For behavior analysis, worms were age-synchronized by transferring 30–50 young adult worms onto an NGM plate seeded with OP50 and allowing them to lay eggs for 2 hr before removal. The plates were then kept at room temperature (22°C) for 3 days, allowing the eggs to develop into young adults, identified by the presence of 5–10 eggs arranged in single rows within the gonads of all worms on the plate. To wash the synchronized adults off the plate, worms were gently transferred into Eppendorf tubes using glass Pasteur pipettes. After allowing the worms to settle at the bottom of the tube (~1 min), the supernatant was removed and replaced with fresh M9 buffer. This washing step was repeated three times to remove residual OP50. Next, approximately 30–70 worms were loaded into an inlet of the chambers shown in *Figure 1* using a 20 μl pipette and allowed to roam freely in the chamber for 60 min before behavior in the assay area was recorded.

To assay worms confined to the assay area by a barrier (*Figure 1—figure supplement 1A*), PDMS chambers were prepared with a removable gap separating the training and assay zones. M9 buffer was first injected into the exploration zone, after which ~100 worms were loaded and allowed to explore for 10 min, during which they moved only within the reach of the M9 buffer, typically up to the gap between the exploration and assay zones. To remove the barrier, additional M9 buffer was gently introduced into the assay zone until it merged with the buffer, allowing worms to move freely between zones. After 15 min of exploration, the barrier was reintroduced by completely drying the space between the zones with a Kimwipe, once again confining worms to areas containing M9 buffer. Behavior in the assay zone was recorded immediately after the barrier was reinserted and again 40 min later.

## Fluorescence microscopy

Widefield fluorescence microscopy was performed using a Leica DMi8 inverted microscope equipped with 63x/1.4NA and 100x/1.4NA oil-immersion objectives (Leica Biosystems, NJ). Worms were mounted on thin 3% agar pads and anesthetized with 10 mM levamisole diluted in M9 buffer. Images were acquired with an Andor iXon-888 EMCCD camera (Oxford Instruments, England), with an exposure time of 150 ms and an electron-multiplying gain of 200. Acquisition settings were kept constant across experiments.

## Statistical analysis

Behavior measurements were calculated for each individual chamber, and these chamber-specific values were then used to generate the final mean values shown in the bar plots. The sample size (approximately 25) was defined as the number of chambers analyzed per experiment and was determined by power analysis using wild-type data. An unpaired Student's *t*-test with Bonferroni correction was used to compare average speeds between uniform and binary chambers. To quantify the relative difference between groups, we calculated the percent change for each value in the uniform chambers relative to the mean of value from the binary chambers. Specifically, we subtracted the mean of the values from the binary chamber from each individual value in the uniform chamber. The result was then divided by the mean of the binary chamber and multiplied by 100 to express the difference as a percentage. For comparisons among multiple strains, we conducted one-way ANOVA followed by a Tukey–Kramer post hoc test on all possible pairwise comparisons. All statistical analyses were performed using R.

## Acknowledgements

This research was supported by NIH grants R01NS109476 and R01NS115974 to JB. MR was supported by T32GM136534 and F31NS129545. The authors thank the *Caenorhabditis elegans* Genetics Center (funded by the NIH Office of Research Infrastructure Programs P40 OD010440) for providing worm strains. We also thank Lin Zhang and Yan Liu for their support in constructing DNA plasmids and generating transgenic worm strains. Special thanks to Melissa Woo (Mercer Island High School), Tabitha Ngo, and Sahithi Challam (Interlake High School) for their assistance with worm maintenance. We appreciate the members of the Bai lab for their critical reading of the manuscript. Additionally, we acknowledge the multiple mentorship programs at Fred Hutchinson Cancer Center, including the SURP Undergraduate Researchers and Pathways Undergraduate Researchers programs, supported in part by the Cancer Center Support Grant P30 CA015704, for fostering relationships with high school and undergraduate trainees who contributed to this project.

## Additional information

### Funding

| Funder | Grant reference number | Author |
|---|---|---|
| National Institute of Neurological Disorders and Stroke | R01NS109476 | Jihong Bai |
| National Institute of Neurological Disorders and Stroke | R01NS115974 | Jihong Bai |
| National Institute of General Medical Sciences | T32GM136534 | Manuel Rosero |
| National Institute of Neurological Disorders and Stroke | F31NS129545 | Manuel Rosero |

The funders had no role in study design, data collection, and interpretation, or the decision to submit the work for publication.

### Author contributions

Manuel Rosero, Conceptualization, Data curation, Formal analysis, Investigation, Visualization, Methodology, Writing - original draft, Writing – review and editing; Jihong Bai, Conceptualization, Resources, Supervision, Funding acquisition, Project administration, Writing – review and editing

### Author ORCIDs

Manuel Rosero ⓘ https://orcid.org/0000-0001-8950-7251
Jihong Bai ⓘ https://orcid.org/0000-0001-6773-2175

Reviewer #1 (Public review): https://doi.org/10.7554/eLife.106496.3.sa1
Reviewer #2 (Public review): https://doi.org/10.7554/eLife.106496.3.sa2
Reviewer #3 (Public review): https://doi.org/10.7554/eLife.106496.3.sa3
Author response https://doi.org/10.7554/eLife.106496.3.sa4

## Additional files

### Supplementary files

MDAR checklist

### Data availability

The data supporting the findings of this study are available in Dryad at https://doi.org/10.5061/dryad.k3j9kd5pz.

The following dataset was generated:

| Author(s) | Year | Dataset title | Dataset URL | Database and Identifier |
|---|---|---|---|---|
| Rosero M, Bai J | 2026 | AFD thermosensory neurons mediate tactile-dependent locomotion modulation in *C. elegans* | https://doi.org/10.5061/dryad.k3j9kd5pz | Dryad Digital Repository, 10.5061/dryad.k3j9kd5pz |

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

## Appendix 1—key resources table

| Reagent type (species) or resource | Designation | Source or reference | Identifiers | Additional information |
|---|---|---|---|---|
| Gene (*Drosophila melanogaster*) | CX2065: *odr-1(n1936) X* | Caenorhabditis Genetics Center | RRID:WB-STRAIN:WBStrain00005220 | |
| Strain, strain background (*Escherichia coli*) | | | | |
| Genetic reagent (*C. elegans*) | VC3024: *gcy-2(ok3721) II* | Caenorhabditis Genetics Center | RRID:WB-STRAIN:WBStrain00037628 | |
| Genetic reagent (*C. elegans*) | VC2796: *gcy-3(gk1154) II* | Caenorhabditis Genetics Center | RRID:WB-STRAIN:WBStrain00037548 | |
| Genetic reagent (*C. elegans*) | VC20038: *gcy-4(gk152604) II* | Caenorhabditis Genetics Center | RRID:WB-STRAIN:WBStrain00038010 | |
| Genetic reagent (*C. elegans*) | VC20321: *gcy-6(gk245152) V* | Caenorhabditis Genetics Center | RRID:WB-STRAIN:WBStrain00038258 | |
| Genetic reagent (*C. elegans*) | RB1000: *gcy-5(ok921) II* | Caenorhabditis Genetics Center | RRID:WB-STRAIN:WBStrain00031709 | |
| Genetic reagent (*C. elegans*) | VC20321: *gcy-6(gk245152) V* | Caenorhabditis Genetics Center | RRID:WB-STRAIN:WBStrain00038258 | |
| Genetic reagent (*C. elegans*) | VC20451: *gcy-12(gk142661) II* | Caenorhabditis Genetics Center | RRID:WB-STRAIN:WBStrain00038385 | |
| Genetic reagent (*C. elegans*) | VC3242: *gcy-13(gk3189) V* | Caenorhabditis Genetics Center | RRID:WB-STRAIN:WBStrain00037738 | |
| Genetic reagent (*C. elegans*) | JN1194: *gcy-14(pe1102) V* | Caenorhabditis Genetics Center | RRID:WB-STRAIN:WBStrain00022697 | |
| Genetic reagent (*C. elegans*) | VC2675: *gcy-15(gk1102) II* | Caenorhabditis Genetics Center | RRID:WB-STRAIN:WBStrain00037490 | |
| Genetic reagent (*C. elegans*) | VC2450: *gcy-17(gk1155) I* | Caenorhabditis Genetics Center | RRID:WB-STRAIN:WBStrain00037338 | |
| Genetic reagent (*C. elegans*) | VC30137: *gcy-18(gk423024) IV* | Caenorhabditis Genetics Center | RRID:WB-STRAIN:WBStrain00038860 | |
| Genetic reagent (*C. elegans*) | RB1935: *gcy-20(ok2538) V* | Caenorhabditis Genetics Center | RRID:WB-STRAIN:WBStrain00032620 | |
| Genetic reagent (*C. elegans*) | VC40920: *gcy-21(gk882422) II* | Caenorhabditis Genetics Center | RRID:WB-STRAIN:WBStrain00039885 | |
| Genetic reagent (*C. elegans*) | RB924: *gcy-23(ok797) IV* | Caenorhabditis Genetics Center | RRID:WB-STRAIN:WBStrain00031636 | |
| Genetic reagent (*C. elegans*) | VC2375: *gcy-25(gk1187) IV* | Caenorhabditis Genetics Center | *RRID:WB-STRAIN:WBStrain00037293* | |
| Genetic reagent (*C. elegans*) | RB2622: *gcy-27(ok3653) IV* | Caenorhabditis Genetics Center | RRID:WB-STRAIN:WBStrain00033296 | |
| Genetic reagent (*C. elegans*) | VC20193: *gcy-29(gk162111) II* | Caenorhabditis Genetics Center | RRID:WB-STRAIN:WBStrain00038135 | |
| Genetic reagent (*C. elegans*) | CZ3714: *gcy-31(ok296) X* | Caenorhabditis Genetics Center | RRID:WB-STRAIN:WBStrain00005387 | |
| Genetic reagent (*C. elegans*) | RB1048: *gcy-32(ok995) V* | Caenorhabditis Genetics Center | RRID:WB-STRAIN:WBStrain00031755 | |
| Genetic reagent (*C. elegans*) | CZ3715: *gcy-33(ok232) V* | Caenorhabditis Genetics Center | RRID:WB-STRAIN:WBStrain00005388 | |
| Genetic reagent (*C. elegans*) | CX6448: *gcy-35(ok769) I* | Caenorhabditis Genetics Center | RRID:WB-STRAIN:WBStrain00005280 | |

*Appendix 1—key resources table continued on next page*

*Appendix 1—key resources table continued*

| Reagent type (species) or resource | Designation | Source or reference | Identifiers | Additional information |
|---|---|---|---|---|
| Genetic reagent (C. elegans) | AX1297: gcy-36(db66) X | Caenorhabditis Genetics Center | RRID:WB-STRAIN:WBStrain00000306 | |
| Genetic reagent (C. elegans) | RB626: gcy-37(ok384) IV | Caenorhabditis Genetics Center | RRID:WB-STRAIN:WBStrain00031376 | |
| Genetic reagent (C. elegans) | PR694: tax-2(p694) I | Caenorhabditis Genetics Center | RRID:WB-STRAIN:WBStrain00030790 | |
| Genetic reagent (C. elegans) | PR678: tax-4(p678) III | Caenorhabditis Genetics Center | RRID:WB-STRAIN:WBStrain00030785 | |
| Genetic reagent (C. elegans) | FX31482: cng-2(tm4267) IV | National BioResource Project, Japan | WBStrain00063972 | |
| Genetic reagent (C. elegans) | VC40261: cng-3(gk541751) IV | Caenorhabditis Genetics Center | RRID:WB-STRAIN:WBStrain00039244 | |
| Genetic reagent (C. elegans) | VC20290: cng-4(gk195496) IV | Caenorhabditis Genetics Center | RRID:WB-STRAIN:WBStrain00038229 | |
| Genetic reagent (C. elegans) | IK800: gcy-8(oy44) IV | Caenorhabditis Genetics Center | RRID:WB-STRAIN:WBStrain00022015 | |
| Genetic reagent (C. elegans) | IK429: gcy-18(nj38) IV | Caenorhabditis Genetics Center | RRID:WB-STRAIN:WBStrain00021997 | |
| Genetic reagent (C. elegans) | BJH2598: cng-3(gk541751) IV; ttTi5605 II | This study | WBStrain00063958 | This strain can be obtained by contacting the corresponding author. |
| Genetic reagent (C. elegans) | BJH2596: gcy-18(gk423024) IV; ttTi5605 II | This study | WBStrain00063959 | This strain can be obtained by contacting the corresponding author. |
| Genetic reagent (C. elegans) | BJH3481: pekSi609[gcy-18p::gcy-18::sl2::mNeonGreen::rab-3 3'utr; vha-6p::mScarlet; hygR(+)] ttTi5605 II; gcy-18(gk423024) IV | This study | WBStrain00063960 | This strain can be obtained by contacting the corresponding author. |
| Genetic reagent (C. elegans) | BJH3527: pekSi655[srtx-1Bp::cng-3::sl2::mNeonGreen::rab-3 3'utr; vha-6p::mScarlet; hygR(+)] ttTi5605 II; cng-3(gk541751) IV | This study | WBStrain00063961 | This strain can be obtained by contacting the corresponding author. |
| Genetic reagent (C. elegans) | BJH3539: pekSi667[gcy-18p::cng-3::sl2::mNeonGreen::unc-54 3'utr; vha-6p::mScarlet; hygR(+)] ttTi5605 II; cng-3(gk541751) IV | This study | WBStrain00063962 | This strain can be obtained by contacting the corresponding author. |
| Genetic reagent (C. elegans) | GN112: pgIs2[gcy-8p::TU#813; gcy-8p::TU#814; unc-122p::gfp; gcy-8p::mCherry; gcy-8p::gfp; ttx-3p::gfp] | Caenorhabditis Genetics Center | RRID:WB-STRAIN:WBStrain00007871 | |
| Genetic reagent (C. elegans) | LX1024: kcc-3(ok228) II | Gift from the Singhvi Lab | WBStrain00063971 | This strain can be obtained from Singhvi lab. |

*Appendix 1—key resources table continued on next page*

*Appendix 1—key resources table continued*

| Reagent type (species) or resource | Designation | Source or reference | Identifiers | Additional information |
|---|---|---|---|---|
| Genetic reagent (C. elegans) | PR767: ttx-1(p767) V | Caenorhabditis Genetics Center | RRID:WB-STRAIN:WBStrain00030792 | |
| Genetic reagent (C. elegans) | JN578: peIs578 [npr-9p::casp1; npr-9p::Venus; unc-122p::mCherry] | Caenorhabditis Genetics Center | RRID:WB-STRAIN:WBStrain00051726 | |
| Genetic reagent (C. elegans) | ZB2551: mec-10 (tm1552) X | Caenorhabditis Genetics Center | RRID:WB-STRAIN:WBStrain00040801 | |
| Genetic reagent (C. elegans) | BJH976: pekSi102[egl-44p::mec-10::unc-54 3'utr; neoR(+)] ttTi5605 II; mec-10(tm1552) X. | This study | WBStrain00063954 | This strain can be obtained by contacting the corresponding author. |
| Genetic reagent (C. elegans) | BJH977: pekSi103[mec-18p::mec-10::unc-54 3'utr; neoR(+)] ttTi5605 II; mec-10(tm1552) X. | This study | WBStrain00063955 | This strain can be obtained by contacting the corresponding author. |
| Genetic reagent (C. elegans) | BJH978: pekSi104[mec-10p::mec-10::unc-54 3'utr; neoR(+)] ttTi5605 II; mec-10(tm1552) X. | This study | WBStrain00063956 | This strain can be obtained by contacting the corresponding author. |
| Genetic reagent (C. elegans) | BJH4059: pekSi102[egl-44p::mec-10::unc-54 3'utr; neoR(+)] ttTi5605 II; pekSi715[mec-18p::mec-10::unc-54 3'utr; neoR(+)] cxTi10816 IV; mec-10(tm1552) X | This study | WBStrain00063957 | This strain can be obtained by contacting the corresponding author. |
| Genetic reagent (C. elegans) | FG927: inx-1(tm6662) X. | Caenorhabditis Genetics Center | RRID:WB-STRAIN:WBStrain00054739 | |
| Genetic reagent (C. elegans) | FG0614: inx-4 (ok2373) V | Gift from the Ferkey lab | WBStrain00063970 | This strain can be obtained from Ferkey lab. |
| Genetic reagent (C. elegans) | RB1792: inx-7(ok2319) IV | Caenorhabditis Genetics Center | RRID:WB-STRAIN:WBStrain00032483 | |
| Genetic reagent (C. elegans) | RB2051: inx-10(ok2714) V | Caenorhabditis Genetics Center | RRID:WB-STRAIN:WBStrain00032735 | |
| Genetic reagent (C. elegans) | FX01896: inx-19 (tm1896) I | National BioResource Project, Japan | WBStrain00063969 | |
| Genetic reagent (C. elegans) | BJH2919: inx-7(ok2319) IV; inx-10(ok2714) V | This study | WBStrain00063963 | This strain can be obtained by contacting the corresponding author. |
| Genetic reagent (C. elegans) | BJH1162: pekEx302 [srtx-1Bp::cx36::sl2::mNeonGreen; inx-1p::cx36::sl2::mKate; Cbr-unc-119(+); unc-122p::gfp; neoR(+)]; unc-119(ed3) III; inx-7(ok2319) IV; inx-10(ok2714) V | This study | WBStrain00063964 | This strain can be obtained by contacting the corresponding author. |

*Appendix 1—key resources table continued on next page*

*Appendix 1—key resources table continued*

| Reagent type (species) or resource | Designation | Source or reference | Identifiers | Additional information |
|---|---|---|---|---|
| Genetic reagent (C. elegans) | BJH4060: pekSi717[srtx-1Bp::inx-10a(genomic)::sl2::gfp::let-858 3'UTR; vha-6p::mScarlet; HygR(+)] ttTi5605 II; inx-10(ok2714) V | This study | WBStrain00063965 | This strain can be obtained by contacting the corresponding author. |
| Genetic reagent (C. elegans) | BJH833: muIs253[eft-3p::gfp1-10::unc-54 3'utr; Cbr-unc-119(+)]; unc-119(ed3) III; gcy-18(pek331[gcy-18::gfp11x7]) IV | This study | WBStrain00063966 | This strain can be obtained by contacting the corresponding author. |
| Genetic reagent (C. elegans) | BJH4056: muIs253[eft-3p::gfp1-10::unc-54 3'utr, Cbr-unc-119(+)]; kcc-3(ok228) II; gcy-18(pek331[gcy-18::gfp11x7]) IV | This study | WBStrain00063968 | This strain can be obtained by contacting the corresponding author. |
| Genetic reagent (C. elegans) | BJH4057: muIs253[eft-3p::gfp1-10::unc-54 3'utr, Cbr-unc-119(+)]; gcy-18(pek331[gcy-18::gfp11x7]) IV; ttx-1(p767) V | This study | WBStrain00063967 | This strain can be obtained by contacting the corresponding author. |
| Recombinant DNA reagent | BJP-MR16: gcy-18p::gcy-18::sl2::mNeonGreen::rab-3 3'utr; vha-6p::mScarlet::tbb-2 3'utr; hygR(+) | This study | | This plasmid can be obtained by contacting the corresponding author. |
| Recombinant DNA reagent | BJP-MR65: srtx-1Bp::cng-3::sl2::mNeonGreen::rab-3 3'utr; vha-6p::mScarlet::tbb-2 3'utr; hygR(+) | This study | | This plasmid can be obtained by contacting the corresponding author. |
| Recombinant DNA reagent | BJP-MR93: gcy-18p::cng-3::sl2::mNeonGreen::rab-3 3'utr; vha-6p::mScarlet::tbb-2 3'utr; hygR(+) | This study | | This plasmid can be obtained by contacting the corresponding author. |
| Recombinant DNA reagent | BJP-MR96: inx-1p::cx36::sl2::mKate::rab-3 3'utr | This study | | This plasmid can be obtained by contacting the corresponding author. |
| Recombinant DNA reagent | BJP-MR97: srtx-1bp::cx36::sl2::mKate::rab-3 3'utr | This study | | This plasmid can be obtained by contacting the corresponding author. |
| Recombinant DNA reagent | BJP-A433: mec-18p::mec-10::unc-54 3'utr; neoR(+) | This study | | This plasmid can be obtained by contacting the corresponding author. |

*Appendix 1—key resources table continued on next page*

*Appendix 1—key resources table continued*

| Reagent type (species) or resource | Designation | Source or reference | Identifiers | Additional information |
|---|---|---|---|---|
| Recombinant DNA reagent | BJP-A433: *egl-44p::mec-10::unc-54 3'utr; neoR(+)* | This study | | This plasmid can be obtained by contacting the corresponding author. |
| Commercial assay, kit | SuperScript III Reverse Transcriptase kit | Thermo Fisher Scientific, MA | Cat# 18080044 | |
| Commercial assay, kit | MultiSite Gateway Construction Kit | Thermo Fisher Scientific, MA | Cat# 12537100 | |
| Commercial assay, kit | Alt-R CRISPR-Cas9 tracrRNA | Integrated DNA Technologies (IDT, NJ) | Cat# 1072532 | |
| Commercial assay, kit | Alt-R S.p. Cas9 Nuclease V3 | Integrated DNA Technologies (IDT, NJ) | Cat# 1081058 | |
| Sequence-based reagent | gcy-18_crRNA | Integrated DNA Technologies (IDT, NJ) | Alt-R CRISPR-Cas9 crRNA | TACAACGACTGAAAGAGGAG |
| Commercial assay, kit | Sylgar 184 Silicone Elastomer Kit | Dow Corning, MI | Material# 4019862 | |
| Software, algorithm | WormLab | MBF Bioscience, VT | | |

