## [Editor Report · eLife Assessment]

The manuscript presents **important** findings on how *C. elegans* can utilize distinct molecular mechanisms and circuit engagements to regulate tactile-dependent locomotory behaviours through the AFD thermosensory neuron. The authors use multiple techniques including microfluidics, genetic manipulations and single-copy rescue experiments, to provide **compelling** evidence for the role of AFD/AIB electrical synaptic connections in this behaviour. The reviewers are satisfied with the comprehensive revisions made by the authors.

---

## [Referee Report · Reviewer #1 (Public review)]

Summary:

In this manuscript, Rosero and Bai examined how the well-known thermosensory neuron in *C. elegans*, AFD, regulates context-dependent locomotory behavior based on the tactile experience. Here they show that AFD uses discrete cGMP signalling molecules and independent of its dendritic sensory endings regulates this locomotory behavior. The authors also show here that AFD's connection to one of the hub interneurons, AIB, through gap junction/electrical synapses, is necessary and sufficient for the regulation of this context-dependent locomotion modulation.

Strengths:

This is an interesting paper showcasing how a sensory neuron in *C. elegans* can employ a distinct set of molecular strategies and different physical parts to regulate a completely distinct set of behaviors, which were not been shown to be regulated by AFD before. The experiments were well performed and the results are clear. However, there are some questions about the mechanism of this regulation. This reviewer thinks that the authors should address these concerns before the final published version of this manuscript.

Comments on revisions:

In this revised manuscript, Rosero and Bai satisfactorily addressed all the concerns raised by this reviewer regarding their original manuscript. This reviewer appreciates the authors' effort. This revised and improved manuscript demonstrates that a sensory neuron in *C. elegans* can utilize distinct molecular strategies and circuit engagements to regulate distinct sets of behaviors. This reviewer believes that the manuscript is suitable for final acceptance in eLife.

---

## [Referee Report · Reviewer #2 (Public review)]

The goal of the study was to uncover the mechanisms mediating tactile-context-dependent locomotion modulation in *C. elegans*, which represents an interesting model of behavioral plasticity. Starting from a candidate genetic screen focusing on guanylyl cyclase (GCY) mutants, the authors identified the AFD-specific gcy-18 gene as essential for tactile-context-dependent locomotion modulation. AFD has been primarily characterized as a thermosensory neuron. However, key thermosensory transduction genes and the sensory ending structure of AFD were shown here to be dispensable for tactile-context locomotion modulation. AFD actuates tactile-context locomotion modulation via the cell-autonomous actions of GCY-18 and the CNG-3 cyclic nucleotide-gated channel, and via AFD's connection with AIB interneurons through electrical synapses. At the circuit level, AIB also receive inputs from the mechanosensory neuron FLP, which was also shown to be relevant for tactile-context-dependent locomotion modulation.

For this study, the authors combined a very clever microfluidic-based behavioral assay with a large set of genetic manipulations to dissect the molecular and cellular pathways involved. Rescue experiments with single-copy transgenes are particularly convincing. The study is very clearly written, and the figures are nicely illustrated with diagrams that effectively convey the authors' interpretation. Overall, the convergence of behavioral assays, genetics, and circuit analysis provides convincing support for the proposed role of the AFD-AIB connection, potentially downstream of FLP via synapic and of other mechanosensory neurons via extra-synaptic communication.

The facts that AFD mediates tactile-context locomotion modulation, that this role relies on GCY-18, and on electrical synapses linking AFD to AIB are new, somewhat unexpected, and interesting. The study raises intriguing and addressable questions about the role of innexin-based cellular communication in a multimodal sensory-behavior microcircuit, including the direction and nature of the signal(s) transmitted through these electrical synapses. These questions remain difficult to address in most experimental systems. The compact and genetically tractable nervous system of *C. elegans* provides a powerful entry point for addressing them in the context of an intact in vivo circuit.

---

## [Referee Report · Reviewer #3 (Public review)]

Summary:

Rosero and Bai report an unconventional role of AFD neurons in mediating tactile-dependent locomotion modulation, independent of their well-established thermosensory function. They partially elucidate the signaling mechanisms underlying this AFD-dependent behavioral modulation. The regulation does not require the sensory dendritic endings of AFD but rather the AFD neurons themselves. This process involves a distinct set of cGMP signaling proteins and CNG channel subunits separate from those involved in thermosensation or thermotaxis. Furthermore, the authors demonstrate that AIB interneurons connect AFD to mechanosensory circuits through electrical synapses. They conclude that, beyond its primary function in thermosensation, AFD contributes to context-dependent neuroplasticity and behavioral modulation via broader circuit connectivity.

While the discovery of multifunctionality in AFD is not entirely unexpected, given the limited number of neurons in *C. elegans* (302 in total), the molecular and cellular mechanisms underlying this AFD-dependent behavioral modulation, as revealed in this study, provide valuable insights into the field.

Strengths:

(1) The authors uncover a novel role of AFD neurons in mediating tactile-dependent locomotion modulation, distinct from their well-established thermosensory function, providing an important conceptual contribution to our understanding of how individual neurons can support multiple, mechanistically separable behavioral functions.

(2) They provide meaningful mechanistic insight into how AFD, GCY-18-dependent cGMP signaling, and AFD-AIB electrical coupling contribute to this AFD-dependent behavioral modulation.

(3) The neural behavior assays utilizing two types of microfluidic chambers (uniform and binary chambers) are innovative and well-designed. In the revised manuscript the authors introduce a removable-barrier assay that physically separates exploration and assay phases. This independent behavioral approach addresses prior concerns about ongoing sensory input and confirms that tactile experience alone is sufficient to modulate locomotion.

(4) By comparing AFD's role in locomotion modulation to its thermosensory function throughout the study, the authors present strong evidence supporting these as two independent functions of AFD.

(5) The finding that AFD contributes to context-dependent behavioral modulation is significant, further reinforcing the growing evidence that individual neurons can serve multiple functions through broader circuit connectivity.

Weaknesses:

While the requirement for AFD, GCY-18, and AFD-AIB electrical coupling is well supported, the directionality of information flow and the precise mode of interaction between mechanosensory neurons, AIB, and AFD remain unclear and an area of future studies.

Overall, the authors successfully achieve their primary aim of identifying and characterizing a novel role for AFD in tactile experience-dependent locomotion modulation. This work contributes meaningfully to the growing body of literature demonstrating multifunctionality and context-dependent reconfiguration of individual neurons within compact nervous systems.

---

## [Author Response]

The following is the authors’ response to the original reviews.

**Public Reviews:**
Although the reviewers agree on the potential importance of this study, they have brought out multiple pertinent queries with respect to the interpretation of some of the results presented in the manuscript, that the authors should consider addressing. The reviewers have also suggested modifications that would increase the clarity of the manuscript.

We appreciate the thoughtful evaluation of our manuscript by the reviewers and the editor. We are encouraged by their recognition of the importance of our study and have carefully considered all the points raised. In response, we have added new data and revised the text to address the concerns and improve the clarity of the manuscript. Our detailed responses to the reviewers’ comments are provided below.

**Reviewer #1 (Public review):**
Summary:In this manuscript, Rosero and Bai examined how the well-known thermosensory neuron in *C. elegans*, AFD, regulates context-dependent locomotory behavior based on the tactile experience. Here they show that AFD uses discrete cGMP signaling molecules and independent of its dendritic sensory endings regulates this locomotory behavior. The authors also show here that AFD's connection to one of the hub interneurons, AIB, through gap junction/electrical synapses, is necessary and sufficient for the regulation of this context-dependent locomotion modulation.Strengths:This is an interesting paper showcasing how a sensory neuron in *C. elegans* can employ a distinct set of molecular strategies and different physical parts to regulate a completely distinct set of behaviors, which were not been shown to be regulated by AFD before. The experiments were well performed and the results are clear. However, there are some questions about the mechanism of this regulation. This reviewer thinks that the authors should address these concerns before the final published version of this manuscript.Weaknesses:(1) The authors argued about the role of prior exposure to different physical contexts which might be responsible for the difference in their locomotory behavior. However, the worms in the binary chamber (with both non-uniformly sized and spaced pillars) experienced both sets of pillars for one hour prior to the assay and they were also free to move between two sets of environments during the assay. So, this is not completely a switch between two different types of tactile barriers (or not completely restricted to prior experience), but rather a difference between experiencing a more complex environment vs a simple uniform environment. They should rephrase their findings. To strictly argue about the prior experience, the authors need to somehow restrict the worms from entering the uniform assay zone during the 1hr training period.

We agree that, in the original design, worms in the binary chamber experience a more complex physical environment while retaining access to both exploration and assay zones. We have therefore revised the manuscript to more clearly distinguish between behavioral differences due to exposure to a complex environment and modulation driven by prior experience.

To directly test whether locomotion modulation can be sustained by prior physical experience in the absence of continued access to the exploration zone, we introduced a barrier-based assay that prevents worms from re-entering the exploration zone before locomotion is measured. The results section has been revised accordingly to explicitly address this point.

Revisions to the manuscript:

Lines 122-139: Added two paragraphs describing the new assay and summarizing the corresponding results.

“Because worms in the binary chamber are exposed to both pillar types and remain free to move between exploration and assay zones, the behavioral differences described above could reflect exposure to a more complex physical environment rather than prior experience alone. To directly test whether locomotion is modulated by prior physical experience independently of continued access to the exploration zone, we designed microfluidic chambers in which the assay zone could be separated from the exploration zone by a removable barrier (Fig. 1–Supplement 1A). In these chambers, worms were initially allowed to explore the entire device, including exploration zones that either matched or differed from the assay zone. A barrier was then inserted to prevent worms in the assay zone from re-entering the exploration zones.

Under these conditions, locomotion immediately after barrier insertion was higher in worms that had previously explored physical settings matching the assay zone (205 ± 8 µm/s) than in worms that had explored non-matching settings (151 ± 7 µm/s; p = 0.006; Fig. 1–Supplement 1B). This difference persisted when worms were recorded 40 minutes after barrier insertion, with animals in matching chamber retaining their higher locomotion rates (218 ± 11 µm/s) compared to those in non-matching chambers (185 ± 8 µm/s; p = 0.02; Fig. 1–Supplement 1B). These findings demonstrate that prior exploration of distinct physical environments can modulate locomotion even when worms are prevented from returning to those environments, supporting a role for prior physical experience independent of ongoing sensory input.”

Figure 1–Supplement 1: New figure showing the experimental design and behavioral results.

(2) The authors here argued that the sensory endings of AFD are not required for this novel role of AFD in context-dependent locomotion modulation. However, gcy-18 has been shown to be exclusively localized to the ciliated sensory endings of AFD and even misexpression of GCY-18 in other sensory neurons also leads to localizations in sensory endings (Nguyen et. al., 2014 and Takeishi et. al., 2016). They should check whether gcy-18 or tax-2 gets mislocalized in kcc-3 or tax-1 mutants.

As the reviewer suggested, we examined GCY-18 localization in wild type animals and in mutants with defective sensory microvilli using a split-GFP strategy (He et al., 2019). We generated a *gcy18::gfp11×7* knock-in strain using CRISPR–Cas9 to visualize endogenous GCY-18 localization. Consistent with prior studies, GCY-18 localized strongly to the AFD dendritic ending in wild-type animals (Figure 4– Supplement 1A, A′, A′′), with an additional weaker signal detectable near the soma and axon (Figure 4– Supplement 1A′′′).

In *kcc-3* mutants, GCY-18 remained localized to the distal dendrite despite disruption of sensory microvillar morphology (Figure 4–Supplement 1B–B′′). Similarly, in *ttx-1* mutants, which completely lack AFD sensory microvilli, GCY-18 still localized to the distal dendrite (Figure 4–Supplement 1C–C′′) and remained detectable near the soma and axon (Figure 4–Supplement 1C′′′).

In the revised manuscript, we clarify both the implications and the limitations of these imaging experiments, noting that “although these experiments do not identify the precise subcellular site at which GCY-18 acts, they show that disruption of sensory microvilli does not substantially alter GCY-18 localization within AFD.” The exact site at which GCY-18 functions to support locomotion modulation therefore remains an important open question for future investigation.

Revisions to the manuscript:

Figure 4-Supplement 1: Added a new figure reporting GCY-18 localization in wild type and mutant worms.

Lines 268-280: Added a new paragraph reporting GCY-18 localization in wild type, *kcc-3*, and *ttx-1* mutants and clarifying its relevance to the reviewer’s concern.

“Given that *gcy-18* is required for context-dependent locomotion modulation and that GCY-18 localizes to the distal dendrite of AFD, we next examined how disruption of sensory microvilli affects its localization in AFD. We used a split-GFP strategy to visualize endogenous GCY-18 [73]. A tandem array of seven GFP11 β-strands (GFP11x7) was inserted at the C-terminus of GCY-18 using CRISPR-Cas9. When complemented with GFP1-10, GCY-18::GFP11x7 fluorescence was strongly enriched at the AFD sensory microvilli near the nose (Fig. 4–Supplement 1A-A′′), consistent with previous reports [42,74,75]. In addition, weaker but reproducible GCY-18 signal was detected near the AFD soma and axon (Fig. 4–Supplement 1A′′′). Importantly, in *kcc-3*, which exhibit disrupted sensory microvilli, and *ttx-1* mutants, which lack sensory microvilli, GCY-18 remained localized to the distal dendrite and was still detectable near the soma and axon (Fig. 4–Supplement 1B-B′′’ and 1C-C′′′). Although these experiments do not identify the precise subcellular site at which GCY-18 acts, they show that disruption or loss of sensory microvilli does not substantially alter GCY-18 localization within AFD.”

(3) MEC-10 was shown to be required for physical space preference through its action in FLP and not the TRNs (PMID: 28349862). Since FLP is involved in harsh touch sensation while TRNs are involved in gentle touch sensation, which are the neuron types responsible for tactile sensation in the assay arena? Does mec-10 rescue in TRNs rescue the phenotype in the current paper?

We performed cell-specific rescue experiments of *mec-10*. Single-copy expression of *mec-10* cDNA in either FLP neurons alone (*egl-44p*) or TRNs alone (*mec-18p*) did not restore context-dependent locomotion modulation (Fig. 5A). In contrast, co-expression in both FLP and TRNs (*egl-44p::mec-10* + *mec18p::mec-10*), as well as expression from the *mec-10* promoter, rescued the phenotype.

These results indicate that input from multiple *mec-10*-expressing neurons, including both FLP and TRNs, is required for context-dependent locomotion adjustment. This requirement differs from spatial preference behavior, where *mec-10* acts specifically in FLP (Han et al., 2017), suggesting distinct mechanosensory circuits are engaged by different tactile-driven behaviors.

Revisions to the manuscript:

Fig. 5A: Updated to include the cell-specific rescue data.

Lines 317-331: Added a new paragraph describing these findings.

“The *mec-10* gene is expressed in several mechanosensory neurons, including the six touch receptor neurons (TRNs) and the polymodal nociceptors FLP and PVD [77,79]. To determine which neurons are required for tactile-dependent locomotion modulation, we expressed *mec-10* cDNA under cell-specific promoters: *mec-18p* (TRNs) [80], *egl-44p* (FLP) [81], or *mec-10p* (TRNs, FLP, and PVD) [79]. Expression in either FLP or TRNs alone did not restore modulation, as worms carrying *egl-44p::mec-10* (Δspeed: -11± 4%) or *mec-18p::mec-10* (Δspeed: -13 ± 4%) transgenes showed significantly reduced Δspeed compared to wild type (Δ speed: N2: 33 ± 6%; p < 0.0001 for both; Fig. 5A). By contrast, *mec-10* co-expression in both FLP and TRNs (Δspeed: 16 ± 4%), or expression from the *mec-10* promoter (Δspeed: 23 ± 4%), restored Δ speed to wild type levels (p = 0.20 and p = 0.57, respectively; Fig. 5A). These findings indicate that *mec10* expression across multiple mechanosensory neuron types is required for context-dependent locomotion modulation. It is also worth noting that, while both tactile-dependent locomotion modulation and previously reported spatial preference require FLP, only the former depends on TRNs. Together, these findings suggest that distinct subsets of mechanosensory neurons differentially contribute to behaviors shaped by tactile experience.”

(4) The authors mention that the most direct link between TRNs and AFD is through AIB, but as far as I understand, there are no reports to suggest synapses between TRNs and AIB. However, FLP and AIB are connected through both chemical and electrical synapses, which would make more sense as per their mec10 data. (the authors mentioned about the FLP-AIB-AFD circuit in their discussion but talked about TRNs as the sensory modality). mec-10 rescue experiment in TRNs would clarify this ambiguity.

We agree with the reviewer that there are no reported synapses between TRNs and AIB, and we have revised Fig. 5 and the corresponding text to clarify this point. In the revised manuscript, we removed any implication of a direct TRN-AIB connection and instead focus on the established FLP-AIB-AFD pathway, while considering potential indirect contributions from TRNs.

As the reviewer suggested, we performed cell-specific *mec-10* rescue experiments. Expression of *mec-10* in either FLP alone or TRNs alone was insufficient to restore tactile-dependent locomotion modulation, whereas co-expression in both cell types rescued the phenotype (revised Fig. 5A). These results indicate that FLP is essential for this behavior, consistent with the known FLP-AIB-AFD connectivity, and that TRNs are also required.

Given that TRNs lack direct synapses with AIB, TRN requirement suggests the involvement of indirect communication, likely mediated through modulatory mechanisms such as neuropeptide signaling. Accordingly, we have revised the model (revised Fig. 5C) and the corresponding text to clarify that tactiledependent locomotion modulation integrates inputs from multiple *mec-10*-expressing neurons and does not rely on a direct TRN-AIB synaptic connection.

Revisions to the manuscript:

Lines 334–345: Revised paragraph to clarify circuit logic and remove implication of direct TRN-AIB synapses.

“Touch-sensitive neurons that express *mec-10*, including TRNs, FLP, and PVD, do not form direct synapses with AFD, suggesting that tactile information is relayed through intermediary neurons. Because the interneuron AIB receives synaptic input from FLP and forms electrical synapses with AFD, we hypothesized that AIB could serve as a conduit for mechanosensory signals to reach AFD. To test whether AIB is required for tactile-dependent modulation, we examined locomotion in worms with genetically ablated AIB neurons using *npr-9p::caspase* expression [82]. AIB-ablated worms failed to adjust locomotion speed, showing a near-complete loss of modulation (∆speed: -1 ± 5%) compared to wild type (30 ± 8%, p = 0.001, Fig. 5B). These results demonstrate that AIB is required for AFD-mediated tactile-dependent locomotion modulation. However, because *mec-10*-expressing TRNs are also required, additional pathways beyond AIB likely contribute to transmitting tactile information to AFD, potentially involving indirect synaptic connections through other interneurons or long-distance signaling via neuropeptides or other modulators (Fig. 5C).”

Fig. 5: Updated to include new cell-specific *mec-10* rescue data and revised model.

(5) Do inx-7 or inx-10 rescue in AFD and AIB using cell-specific promoters rescue the behavior?

Yes. We tested this during revision. Using the AFD-specific *srtx-1b* promoter, we expressed *inx10* cDNA selectively in AFD neurons of *inx-10* mutant worms. This manipulation significantly restored tactile-dependent locomotion modulation compared to non-transgenic *inx-10* mutants (Fig. 6D), demonstrating that *inx-10* expression in AFD alone is sufficient to rescue the behavioral defect.

Revisions to the manuscript:

Line 366-370: Added a description of the AFD-specific *inx-10* rescue results.

“We next tested whether restoring *inx-10* specifically in AFD would be sufficient to rescue the behavioral defect. Using the AFD-specific *srtx-1b* promoter, we expressed *inx-10* cDNA in *inx-10* mutant worms. These transgenic animals displayed significantly improved locomotion modulation (∆speed: 42 ± 5%) compared to non-transgenic *inx-10* mutants (15 ± 4%; p = 0.018; Fig. 6D), indicating that *inx-10* expression in AFD alone is sufficient to restore function.”

Fig. 6D: Updated to include new cell-specific *inx-10* rescue data.

(6) How Guanylyl cyclase gcy-18 function is related to the electrical synapse activity between AFD and AIB? Is AFD downstream or upstream of AIB in this context?

At present, the precise relationship between GCY-18 signaling and the AFD-AIB electrical synapse is not fully resolved. Given that AIB receives mechanosensory input from FLP, it is likely that AIB acts upstream of AFD during tactile-dependent locomotion modulation. However, because the AIB-AFD connection is mediated by gap junctions, communication could also be bi-directional, especially since small signaling molecules such as cGMP and Ca^2+^ are known to diffuse through electrical synapses.

We have therefore revised the manuscript to state explicitly that the directionality of information flow between AFD and AIB remains open, and that this will be an important question for future investigation (Line 455-458).

“Together, these findings support a model in which AIB functions as a hub neuron that relays mechanosensory input from FLP to AFD to modulate locomotion (Fig. 5C). However, because electrical synapses are often bidirectional, information flow may also occur in the opposite direction, from AFD to AIB.”

**Reviewer #2 (Public review):**
Summary:The goal of the study was to uncover the mechanisms mediating tactile-context-dependent locomotion modulation in *C. elegans*, which represents an interesting model of behavioral plasticity. Starting from a candidate genetic screen focusing on guanylate cyclase (GCY) mutants, the authors identified the AFDspecific gcy-18 gene as essential for tactile-context-dependent locomotion modulation. AFD is primarily characterized as a thermo-sensory neuron. However, key thermosensory transduction genes and the sensory ending structure of AFD were shown here to be dispensable for tactile-context locomotion modulation. AFD actuates tactile-context locomotion modulation via the cell-autonomous actions of GCY-18 and the CNG-3 cyclic nucleotide-gated channel, and via AFD's connection with AIB interneurons through electrical synapses. This represents a potentially relevant synaptic connection linking AFD to the mechanosensory-behavior circuit.Strengths:(1) The fact that AFD mediates tactile-context locomotion modulation is new, rather surprising, and interesting.(2) The authors have combined a very clever microfluidic-based behavioral assay with a large set of genetic manipulations to dissect the molecular and cellular pathways involved. Rescue experiments with singlecopy transgenes are very convincing.(3) The study is very clearly written, and figures are nicely illustrated with diagrams that effectively convey the authors' interpretation.Weaknesses:(1) Whereas GCY-18 in AFD and the AFD-AIB synaptic connection clearly play a role in tactile-context locomotion modulation, whether and how they actually modulate the mechanosensory circuit and/or locomotion circuit remains unclear. The possibility of non-synaptic communication linking mechanosensory neurons and AFD (in either direction) was not explored. Thus, in the end, we have not learned much about what GCY-18 and the AFD-AIB module are doing to actuate tactile context-dependent locomotion modulation.

We agree with the reviewer that although GCY-18 in AFD and the AFD-AIB connection are clearly required for tactile context-dependent locomotion modulation, the precise mechanisms by which they influence mechanosensory and locomotor circuits remain unresolved. In particular, the possibility of nonsynaptic communication or bidirectional signaling between mechanosensory neurons and AFD cannot be addressed by the current experiments and warrants future investigation.

At the same time, we believe this study reveals several previously unrecognized aspects of tactiledependent locomotion modulation that provide a foundation for future mechanistic investigation.

Specifically, we show that (i) GCY-18 functions in AFD to support tactile-dependent locomotion modulation; (ii) the cGMP-gated channel TAX-4, required for thermosensation, is dispensable for this process, whereas CNG-3 is required, revealing functional specialization within AFD; (iii) the interneuron AIB is necessary for this modulation; and (iv) restoring a single electrical connection between AFD and AIB using mammalian Cx36 is sufficient to rescue tactile-dependent modulation in innexin mutants.

Accordingly, we now explicitly state in the revised Discussion that “a limitation of this study is that the directionality and mode of information flow between AFD and AIB remain unresolved, and defining this relationship will be an important goal for future investigation” (Line 472-475).

(2) The authors only focused on speed readout, and we don't know if the many behavioral parameters that are modulated by tactile context are also under the control of AFD-mediated modulation.

We used locomotion speed as the primary behavioral readout because it provides a robust measure for detecting whether behavior is modified by prior tactile experience, rather than to capture the full spectrum of motor outputs. This strategy is often used to assess experience-dependent behavioral plasticity across sensory modalities and enabled us to uncover the unexpected role of AFD in tactile-dependent plasticity.

In the revised manuscript, we expanded our analysis to include additional behavioral parameters. As described in the Results, AFD-ablated worms showed a complete loss of context-dependent modulation not only in speed, but also in idle time and turning frequency, with no detectable differences between uniform and binary chambers (Fig. 4E). These data strengthen the conclusion that AFD broadly supports tactiledependent behavioral modulation rather than selectively affecting a single locomotor parameter.

Revisions to the manuscript:

Fig. 4E: Revised panel to include additional locomotion parameters, including idle time and turning frequency, in wild type and AFD-ablated worms.

Lines 283–285: Expanded the results to describe changes in locomotion speed, idle time, or turning frequency of AFD-ablated mutant worms. “These animals showed no detectable differences between uniform and binary chambers in locomotion speed, idle time, or turning frequency (Fig. 4E).”

(3) The AFD-AIB gap junction reconstruction experiment was conducted in an innexin double mutant background, in which the whole nervous system's functioning might be severely impaired, and its results should be interpreted with this limitation in mind.

We appreciate the reviewer’s concern that the innexin double-mutant background may broadly affect nervous system function, and we agree that loss of innexins is not restricted to the AFD-AIB synapse and could introduce global circuit perturbations.

Importantly, however, the specificity of the rescue is informative. In an *innexin* double-mutant background, where electrical coupling is broadly disrupted, re-establishing a single electrical synapse between AFD and AIB using Cx36 was sufficient to restore tactile-dependent locomotion modulation (Fig. 6D). The ability of a targeted AFD-AIB connection to rescue behavior despite the absence of many other electrical synapses argues against a purely global network defect and instead identifies the AFD-AIB electrical synapse as a critical locus for this modulation.

To further address this concern, we performed an additional rescue experiment in a less perturbed genetic background. In the revised manuscript, we show that AFD-specific expression of *inx-10* rescues locomotion modulation in *inx-10* single mutants (Fig. 6D). Together, these complementary rescue approaches, one restoring endogenous innexin function in AFD and the other reconstituting an electrical synapse using Cx36, support the conclusion that AFD-AIB electrical coupling is sufficient to enable tactile-dependent locomotion modulation, rather than reflecting nonspecific recovery of global circuit function.

Revision to the manuscript:

Fig. 6D and Lines 366-370: Added new data and revised text showing that AFD-specific *inx-10* expression restores tactile-dependent locomotion modulation.

“We next tested whether restoring *inx-10* specifically in AFD would be sufficient to rescue the behavioral defect. Using the AFD-specific *srtx-1b* promoter, we expressed *inx-10* cDNA in *inx-10* mutant worms. These transgenic animals displayed significantly improved locomotion modulation (∆speed: 42 ± 5%) compared to non-transgenic *inx-10* mutants (15 ± 4%; p = 0.018; Fig. 6D), indicating that *inx-10* expression in AFD alone is sufficient to restore function.”

**Reviewer #3 (Public review):**
Summary:Rosero and Bai report an unconventional role of AFD neurons in mediating tactile-dependent locomotion modulation, independent of their well-established thermosensory function. They partially elucidate the signaling mechanisms underlying this AFD-dependent behavioral modulation. The regulation does not require the sensory dendritic endings of AFD but rather the AFD neurons themselves. This process involves a distinct set of cGMP signaling proteins and CNG channel subunits separate from those involved in thermosensation or thermotaxis. Furthermore, the authors demonstrate that AIB interneurons connect AFD to mechanosensory circuits through electrical synapses. They conclude that, beyond its primary function in thermosensation, AFD contributes to context-dependent neuroplasticity and behavioral modulation via broader circuit connectivity.While the discovery of multifunctionality in AFD is not entirely unexpected, given the limited number of neurons in *C. elegans* (302 in total), the molecular and cellular mechanisms underlying this AFD-dependent behavioral modulation, as revealed in this study, provide valuable insights into the field.Strengths:(1) The authors uncover a novel role of AFD neurons in mediating tactile-dependent locomotion modulation, distinct from their well-established thermosensory function.(2) They provide partial insights into the signaling mechanisms underlying this AFD-dependent behavioral modulation.(3) The neural behavior assays utilizing two types of microfluidic chambers (uniform and binary chambers) are innovative and well-designed.(4) By comparing AFD's role in locomotion modulation to its thermosensory function throughout the study, the authors present strong evidence supporting these as two independent functions of AFD.(5) The finding that AFD contributes to context-dependent behavioral modulation is significant, further reinforcing the growing evidence that individual neurons can serve multiple functions through broader circuit connectivity.Weaknesses:(1) Limited Behavioral Assays: The study relies solely on neural behavior assays conducted using two types of microfluidic chambers (uniform and binary chambers) to assess context-dependent locomotion modulation. No additional behavioral assays were performed. To strengthen the conclusions, the authors should validate their findings using an independent method, at the very least by testing AFD-ablated animals and gcy-18 mutants with a second behavioral approach.

The reviewer points out that the original study relied on locomotion assays in two microfluidic environments (uniform and binary chambers) and suggests validation using an independent behavioral approach, particularly for AFD-ablated animals and *gcy-18* mutants.

To address this concern, we developed an independent behavioral assay in which the exploration and assay environments are physically separated by a removable barrier (Figure 1–Supplement 1A). In this design, worms first explored distinct physical settings, after which a barrier was inserted to confine them to an identical assay zone. This approach allowed us to directly test whether context-dependent locomotion modulation can be maintained when worms are prevented from re-entering the exploration environment and must rely solely on prior experience.

Using this assay, we found that wild-type worms that had previously explored environments matching the assay zone moved significantly faster than those that had explored non-matching environments (Figure 1– Supplement 1B-C). These results demonstrate that context-dependent locomotion modulation is retained even when ongoing sensory input from the exploration zone is eliminated, independently validating our original findings using a distinct behavioral paradigm.

Further, using this same assay, we found that locomotion modulation was significantly impaired in both *gcy-18* mutants and AFD-ablated worms (Figure 4–Supplement 2A). Together, these results provide independent behavioral evidence supporting the conclusion that AFD and *gcy-18* are required for contextdependent locomotion modulation.

Revision to the manuscript:

Figure 1–Supplement 1A: Added schematic and results from the removable-barrier assay in wild type animals.

Lines 120-137: Added corresponding Results text describing the new assay and wild-type behavior.

“Because worms in the binary chamber are exposed to both pillar types and remain free to move between exploration and assay zones, the behavioral differences described above could reflect exposure to a more complex physical environment rather than prior experience alone. To directly test whether locomotion is modulated by prior physical experience independently of continued access to the exploration zone, we designed microfluidic chambers in which the assay zone could be separated from the exploration zone by a removable barrier (Fig. 1–Supplement 1A). In these chambers, worms were initially allowed to explore the entire device, including exploration zones that either matched or differed from the assay zone. A barrier was then inserted to prevent worms in the assay zone from re-entering the exploration zones.

Under these conditions, locomotion immediately after barrier insertion was higher in worms that had previously explored physical settings matching the assay zone (205 ± 8 µm/s) than in worms that had explored non-matching settings (151 ± 7 µm/s; p = 0.006; Fig. 1–Supplement 1B). This difference persisted when worms were recorded 40 minutes after barrier insertion, with animals in matching chamber retaining their higher locomotion rates (218 ± 11 µm/s) compared to those in non-matching chambers (185 ± 8 µm/s; p = 0.02; Fig. 1–Supplement 1B). These findings demonstrate that prior exploration of distinct physical environments can modulate locomotion even when worms are prevented from returning to those environments, supporting a role for prior physical experience independent of ongoing sensory input.” Figure 4–Supplement 2A: Added data for *gcy-18* mutants and AFD-ablated worms in the removable barrier assay.

Lines 288-296: Added text describing behavioral defects in *gcy-18* mutants and AFD-ablated worms using the new assay.

“Building on our finding that locomotion modulation can be driven by prior physical experience even after worms are prevented from re-entering the exploration zones, we next tested whether AFD is required for this modulation using chambers in which the exploration and assay zones were separated by a removable barrier (Fig. 1–Supplement 1A). Under these conditions, locomotion modulation was significantly reduced in AFD-ablated worms (∆speed: -AFD = 1 ± 6% vs. N2 = 23 ± 7%; p = 0.036; Fig. 4–Supplement 2A). Similarly, *gcy-18* mutants showed defective locomotion modulation (∆speed: *gcy-18* = -1 ± 8% vs. N2 = 23 ± 7%; p = 0.034; Fig. 4–Supplement 2A). These results indicate that AFD and *gcy-18* are required to generate locomotion modulation in response to recent physical experience, even when continued access to surrounding environments is restricted.”

(2) Clarity in Behavioral Assay Methodology: The methodology for conducting the behavioral assays is unclear. It appears that worms were free to move between the exploration and assay zones, with no control over the duration each worm spent in either zone. This lack of regulation may introduce variability in tactile experience across individuals, potentially affecting the reproducibility and quantitativeness of the method. The authors should clarify whether and how they accounted for this variability.

In the primary assay, worms were allowed to move freely between the exploration and assay zones for one hour, and each animal’s tactile experience depended on its exploratory trajectory. To address the resulting variability, we performed an a *priori* power analysis, which determined that approximately 160 worms distributed across more than 20 chambers per condition were sufficient to obtain reliable populationlevel measurements. This sampling strategy was applied consistently across all experiments. Accordingly, analyses emphasize well-powered population means rather than individual trajectories, ensuring robust and reproducible comparisons despite variability in individual experience.

In addition, as described above, we developed a removable-barrier assay that eliminates variability from ongoing exploration by confining worms to the assay zone after a defined exploration period. The consistency of behavioral effects across both assays further supports the robustness and reproducibility of the approach.

(3) Potential Developmental and Behavioral Confounds in Mutant Analysis: Several neuronal mutant strains were used in this study, yet the effects of these mutations on development and general behavior (e.g., movement ability) were not discussed. Although young adult worms were used for behavioral assays, were they at similar biological ages? To rule out confounding factors, locomotion assays assessing movement ability should be conducted (see reference PMID 25561524).

To address the possibility that behavioral phenotypes in mutant strains arise from developmental defects or impaired general locomotion, we directly measured locomotion speed on agar plates and body length in *gcy-18* mutant and AFD-ablated worms. Neither genotype showed defects in basal locomotion speed or body length compared to wild type animals (Figure 4–Supplement 2B-C), indicating that the observed modulation defects are not explained by impaired development or gross motor ability.

To further control for developmental variability, all behavioral assays were performed using agesynchronized populations. Animals were selected at a defined gravid adult stage, identified by the presence of 5-10 eggs arranged in a single row within the gonad. All mutant strains reached this developmental stage approximately three days after egg laying, comparable to wild type animals.

Revision to the manuscript:

Figure 4–Supplement 2B-C: Added quantification of locomotion speed on agar plates and body length for *gcy-18* mutants and AFD-ablated worms.

Lines 297-304: Added text describing the data presented in Figure 4–Supplement 2B-C.

“Finally, to determine whether the modulation defects observed in *gcy-18* mutants and AFD-ablated worms could be attributed to developmental abnormalities or gross motor impairments, we measured locomotion speed and body length on standard NGM plates. Both day-1 adult AFD-ablated worms (speed: 281 ± 10 µm/s; p = 0.33; body length: 1.12 ± 0.01 mm; p = 0.76) and *gcy-18* mutants (speed: 291 ± 13 µm/s; p = 0.22; body length: 1.15 ± 0.02 mm; p = 0.86) showed locomotion speeds and body lengths comparable to wild type controls (speed: 252 ± 30 µm/s; body length: 1.14 ± 0.02 mm; Fig. 4–Supplement 2B, C). These results indicate that the loss of context-dependent locomotion modulation is not due to developmental defects or gross impairments in locomotion.”

(4) Definition and Baseline Measurements for Locomotion Categories: The finding that tax-4 and kcc-3 contribute to basal locomotion but not to context-dependent locomotion modulation is intriguing. The authors argue that distinct mechanisms regulate these two processes; however, the study does not clearly define the concepts of "basal locomotion" and "context-dependent locomotion," nor does it provide baseline measurements. A clear definition and baseline data are needed to support this conclusion.

We define basal locomotion as the locomotion speed of worms measured in the binary chamber, where wild-type animals consistently exhibit lower locomotion rates. Measurements from the binary chamber therefore serve as the baseline reference for locomotion speed in our microfluidic assays. Context-dependent locomotion modulation is defined as the quantified difference in locomotion speed between worms in uniform chambers and those in binary chambers. These definitions are now stated in:

Lines 199-201: “We examined the locomotion speed of mutant worms in the binary chambers, which we refer to as the basal speed because wild type worms consistently move slowest in this environment.”

Lines 645-46: “Asterisks above horizontal black lines indicate statistically significant differences in basal speed, defined as speed of worms in the binary chamber”

**Recommendations for the authors:**

**Reviewer #1 (Recommendations for the authors):**
The availability of strains has not been mentioned. This should be addressed.

The revised Methods section now includes a complete list of strains used in this study, and we have added a statement indicating that all strains are available upon request.

Minor comment:Figure 1C - it should be Idle, not Idel.

We have corrected the y-axis label in Figure 1C to ‘Idle.’

**Reviewer #2 (Recommendations for the authors):**
This is an interesting and well-written article, which I greatly appreciated reading. There are a few concerns that the authors should address, in my opinion, to provide a more complete and convincing story.Major points:(1) Maybe the material transmitted to me was incomplete, but I did not find the gcy gene screen results. It seems important to present the screen results in full, together with the description of the alleles tested for the 24 gcy genes.

The revised manuscript now includes the complete results of the *gcy* mutant screen in Figure 2– Supplement 1, with the alleles tested for all 24 *gcy* genes listed in Table S1.

(2) I did not find the actual p-values, sample sizes for each condition, or raw data; nor a data availability statement indicating where to retrieve these.

Statistical significance is indicated by asterisks in all figures, with definitions provided in each figure legend (n.s., p > 0.05; *, p < 0.05; **, p < 0.01; ***, p < 0.001). Sample sizes are shown as individual data points in the plots, and we have now added explicit n values to each figure legend for clarity. A Data Availability Statement has also been added to indicate where the raw data can be accessed. Where possible, we have included exact p-values. For analyses using Tukey-Kramer post hoc tests, p-values are reported to four decimal places, reflecting the output limits of the statistical software used.

(3) It is not clear why the authors only quantified animal speed for most of the study. What about idle time, turns, and reversals? This choice limits the reach of the study, as we only partly understand what AFD is doing, notably to explain the phenotype in the preference assay.

Data on idle time, turning frequency, and reversal frequency for wild-type worms are now included in Figure 1F. In addition, we present new data showing that AFD ablation disrupts context-dependent modulation of locomotion speed, idle time, and turning frequency (Figure 4E).

(4) Figure 2D and related text: these conclusions are based on a single mutant analysis. Were the millionmutation project lines outcrossed? It would be much more convincing if more gcy alleles were tested (this should be relatively easy since classical alleles are available at the CGC for gcy-8 and gcy-18).

The million-mutation project lines used in this study were outcrossed prior to analysis. In addition, we confirmed that the observed defects were specifically due to loss of *gcy-18* function by rescuing the phenotype through expression of *gcy-18* cDNA under AFD-specific promoters. This cell-specific rescue shows that the behavioral defects arise from disruption of *gcy-18* rather than from background mutations.

(5) It is hard to interpret the speed phenotype when the authors switch between Delta speed and absolute speed display from one figure to another, or even from one panel to another. If only tax-4 and kcc-3 display a constitutive speed phenotype, then there should be no problem showing the absolute speed data in every panel. This is important to convince the reader that major speed changes in mutants are not biasing the interpretation based on Deltas. Indeed, if some mutants move very fast, there might be a ceiling effect. Conversely, if they move very slowly, there might be a 'sickness' effect. Both effects could prevent seeing a tactile-context-dependent modulation, and the results would need to be interpreted much more carefully. Providing the full view on absolute speed levels would also really help support the whole discussion paragraph about the differential regulation of constitutive versus context-dependent locomotion (from L339 onward).

We focus on ∆speed because it directly quantifies experience-dependent locomotion modulation relative to each strain’s own baseline, making it an appropriate metric for comparing tactile plasticity across genotypes. This approach avoids confounding effects from strain-specific differences in overall locomotion levels.

At the same time, we agree that absolute locomotion speed is important to consider when interpreting behavioral phenotypes. To address this, we added plate-based locomotion speed and body length measurements for two key genotypes that lack modulation, *gcy-18* mutants and AFD-ablated worms (Figure 4–Supplement 2B–C). Both exhibit normal locomotion on agar plates, indicating that their defects in tactiledependent modulation are not due to impaired motor ability or general sickness.

In addition, among the mutants tested in microfluidic chambers, *tax-4* mutants display elevated basal speed yet retain robust context-dependent modulation, indicating that ceiling effects do not limit detection of modulation.

(6) The gap junction expression is a nice experiment. But there is a major limitation that should be stated: the electrical synapse re-construction is made in a double mutant background in which the whole animal circuitry might be severely affected. It might well be that the restoration of behavioral plasticity represents something totally irrelevant to wild-type nervous system functioning. A cell-specific innexin knockout is needed to fully support the relevance of the AFD-AIB connection.

We agree that reconstruction of an electrical synapse in an *innexin* double-mutant background carries the limitation that global circuit function may be broadly affected. To address this concern, we performed an additional rescue experiment in a less perturbed genetic background.

As described above, we show that AFD-specific expression of *inx-10* is sufficient to restore tactiledependent locomotion modulation in *inx-10* single mutants (Fig. 6D). This cell-specific rescue does not rely on a double-mutant background and converges on the same outcome as the Cx36-based electrical synapse reconstruction. Together, these complementary approaches support the conclusion that restoring AFD-AIB coupling is sufficient to enable tactile-dependent locomotion modulation, rather than reflecting nonspecific recovery from global circuit disruption.

(7) How was developmental age controlled? It seems that all genotypes were grown for a fixed duration (72h). Some mutants, like gcy-8, might grow slower. It would be useful to at least provide control data in wildtype animals showing that behavioral performance is similar even in slightly younger animals (covering the developmental age of the youngest mutant).

Developmental age was controlled by strict age synchronization and staging criteria rather than growth duration alone. Worms were synchronized by allowing 40-50 young adults to lay eggs on OP50-seeded NGM plates for two hours, after which adults were removed. Developmental stage was further assessed by gonadal morphology, and only young adult animals with 5-10 eggs arranged in a single row were selected for behavioral assays. Using these criteria, all strains, including mutants, consistently reached the assayed stage approximately three days after egg laying, comparable to wild type animals.

To further address the possibility that subtle developmental differences could influence behavior, we measured locomotion speed on agar plates and body length for genotypes that show defects in contextdependent modulation. *gcy-18* mutants and AFD-ablated worms exhibited normal locomotion rates and body size, indicating that their behavioral phenotypes are unlikely to arise from developmental delay or impaired general motor ability. These control data are now included in the revised manuscript (Figure 4– Supplement 2B–C).

(8) Plasmid construction description is entirely lacking.

Description of plasmid construction has been added to the revised Methods.

Minor points:(1) 'Context-dependent locomotion' should be replaced by 'tactile context-dependent locomotion' or something similar throughout the manuscript when referring to the impact of the pillar environment.

Presently, this phrasing shortcut makes the communication too vague throughout, and even confusing when presenting the result of supplementary Figure 2 (where both thermal and tactile contexts are manipulated).

We appreciate this suggestion and have revised the terminology for clarity where appropriate. Prior to introducing the mechanosensory origin of the modulation (that is, before presenting the *mec-10* data), we retain the broader term “context-dependent modulation” to avoid presupposing a tactile mechanism before it is experimentally established.

(2) L97: Suggested change along the same lines: "prior experience" -> "prior tactile experience".

We have made this change as suggested.

(3) Figure 1A: Would the author consider swapping the order of conditions displayed in this diagram? It would make more sense to have the same left-to-right order in the whole figure with the binary chamber on the left, particularly since the author describes the results considering the binary chamber as the 'reference point'.

The order of chambers in Figure 1A has been revised as suggested, with the binary chamber now shown on the left.

(4) Figure 1C: 'idel' typo in the axis label.

The y-axis label has been updated from “idel” to “idle.”

(5) Without AFD-specific manipulations, the data with tax-4 and tax-2 mutants provide limited information regarding TAX-4 and TAX-2 role in AFD. It should be explicitly mentioned in the Results section that they might work in other neurons.

The revised manuscript now explicitly states that the *tax-2(p694)* allele affects multiple neurons, including BAG, ASE, ADE, and AFD (Lines 421–422).

(6) L220-222: The strict meaning of this sentence implies that one attributes a role to AFD in controlling constitutive locomotion, but none of the presented data directly shows this (both kcc-3 and tax-4 mutant phenotypes could arise from additional neurons, regardless of any perturbation in AFD). This should be corrected.

To avoid implying that AFD directly controls constitutive locomotion, we have removed the sentence in question, “Together, these findings suggest that the role of AFD neurons in modulating context-dependent locomotion is distinct from their thermosensory functions and differs from the mechanisms controlling basal locomotion”, from the revised manuscript.

(7) L328-329: Overstatement. Without AFD-specific manipulation of TAX-2 and TAX-4, the different mutant phenotypes could be due to different cell types, rather than different protein pairs in the channel heteromers. I would recommend addressing this alternative possibility directly in the discussion, rather than focusing only on one (I agree, very cool) possibility.

We have clarified this point in the revised text. We now explicitly note that the *tax-2(p694)* mutation affects *tax-2* expression in multiple neurons (AFD, BAG, ASE, and ADE) (Lines 421–422).

**Reviewer #3 (Recommendations for the authors):**
(1) Clarification of inx Gene Expression Analysis (Lines 270-271): The authors should specify how the expression of inx genes in individual neurons was identified.

The revised manuscript now specifies that innexin expression patterns were identified using the CeNGEN single-cell transcriptomic database (Lines 352–354).

(2) Cx36 Expression in AFD and AIB (Lines 287-288): Further clarification is needed on how Cx36 expression was achieved in AFD and AIB.

We have clarified that Cx36 was expressed specifically in AFD using the *srtx-1b* promoter and in AIB using the *inx-1* promoter, as stated in the main text (Lines 372–373) and the Fig. 6 legend.